# PLUG-AND-PLAY OBJECT-CENTRIC REINFORCEMENT LEARNING VIA MASKING

## ABSTRACT

Deep reinforcement learning agents, trained on raw pixel inputs, often fail to generalize beyond their training environments, relying on spurious correlations and irrelevant background details. To address this issue, object-centric agents have recently emerged. However, they require different representations tailored to the task specifications. Inspired by principles of cognitive science and Occam's Razor, we introduce Object-Centric Attention via Masking (OCCAM), a framework for integrating object-centric abstractions into standard visual neural agents. The idea is to selectively preserves task-relevant entities while filtering out irrelevant visual information, while taking advantage of the object-centric inductive bias. Our empirical evaluations on Atari games demonstrate that OCCAM, conditional on object information being available, significantly improves robustness to novel perturbations while showing similar or improved performance compared to conventional pixel-based RL. These results highlight how representation alone, how object cues are encoded and abstracted, substantially influence generalization behavior and performance overall.

## 1 INTRODUCTION

Human visual processing, akin to the dual-system theory of fast and slow thinking (Kahneman, 2011), operates in two phases: a rapid, automatic process that scans the visual field to detect salient features (Treisman, 1985) and a sequential, focused attention mechanism that extracts complex representations from localized regions (Treisman & Gelade, 1980). This hierarchical approach highlights the role of abstract representations—where key entities and their relationships serve as fundamental building blocks for reasoning and planning (Baars, 1993; 2002; Bengio, 2019; Goyal & Bengio, 2022).

Reinforcement learning (RL), in contrast, has predominantly relied on learning directly from raw pixel inputs without an explicit object extraction step, a paradigm introduced with Deep Q-Networks (DQN; (Mnih et al., 2015)). While this task-agnostic approach has enabled remarkable progress, end-to-end convolutional neural networks (CNNs) are often brittle: they exploit spurious correlations, overfit to training environments, and fail under distribution shifts (Farebrother et al., 2018; Agnew & Domingos, 2021; Yoon et al., 2023; Hermann et al., 2024). A particularly striking example is Pong, where agents often rely on the opponent's paddle position—a shortcut feature—while largely ignoring the ball's trajectory (Delfosse et al., 2024b). This failure mode is not unique to Pong: across Atari games, CNN-based agents exhibit sharp performance drops when evaluated outside their training distribution (Delfosse et al., 2025).

These limitations motivate the use of *structured representations* in RL. Recent work explores object-centric decompositions (Zhao et al., 2021; Bertoin et al., 2022), unsupervised object discovery (Lin et al., 2020; Locatello et al., 2020; Delfosse et al., 2023b; Patil et al., 2024), and symbolic policies (Delfosse et al., 2024b; Kohler et al., 2024; Luo et al., 2024; Marton et al., 2024). Complementary strategies improve robustness at the pixel level, including data augmentation (RAD (Laskin et al., 2020), DrQ (Yarats et al., 2021)), saliency-guided filtering (SGQN; (Bertoin et al., 2022)), and reward-driven masking (MaDi; (Grooten et al., 2024)). Across these directions, a consensus emerges: removing unnecessary visual distractions can help generalization when aligned with task structure, though over-abstraction can also discard critical information.

Like many others, we care about an explicit correspondence to task-relevant entities, as emphasized in object-centric RL (OCRL). However, existing methods are often tied to specific pipelines, i.e., it is difficult, if not impossible, to use them with an arbitrary deep reinforcement learner, or they fail easily, such as semantic vector approaches (Delfosse et al., 2024a; Locatello et al., 2020).

We introduce OCCAM, a framework that bridges object-centric RL ideas with standard pipelines. OCCAM evaluates different levels of abstraction—from *Binary Masks* (foreground occupancy) to *Class Masks* (categorical distinctions) to *Plane Masks* that disentangle object types into dedicated channels—making object structure explicit while remaining usable with standard vision-based RL methods. Rather than requiring specialized relational or slot-based architectures, our results show that these standard convolutional backbones can exploit object information when appropriately encoded. Moreover, they provide strong inductive biases for spatial reasoning, including locality, or robust gradient extraction, which align naturally with spatially-grounded object masks, making them even beneficial over more compact architectures such as semantic vectors.

We are solely interested in evaluating the impact of masking information on the performance of neural policies. Thus, the OCCAM framework operates under the light assumption that object information, such as bounding boxes or segmentation masks, is available. This information can be obtained through various means, including engine annotations, (pretrained) detectors, or rule-based systems. OCCAM investigates how such information should be encoded for neural agents, not how to obtain such information. To create a controlled, noise-free testbed, we employ OCAtari's RAM Extraction Method (REM) (Delfosse et al., 2024a) as a perfect extractor in the Atari environment. The objective of OCCAM is not to devise a novel detector, but rather to comprehend the impact of varying abstraction levels, augmented by object information, on robustness, shortcut learning, and performance.

Specifically, our contributions are as follows:

1. We propose OCCAM, a lightweight, task-agnostic framework that introduces object-centric inductive biases through structured input abstraction.

2. We analyze a diverse set of object-centric representations and empirically demonstrate that OCCAM-based agents can match or exceed the performance of pixel and symbolic baselines, improve robustness to perturbations, and reduce shortcut reliance in key cases such as *Pong*.

3. We characterize the limitations of abstraction, showing how overly simplified inputs can hinder performance in tasks that depend on global spatial structure, and outline the trade-offs between simplicity and spatial expressivity. This puts abstraction as a design axis that must be tuned to the task.

4. We find *Plane Masks* works best as a plug-and-play approach across all masks tested.

We proceed as follows. We start off by introducing the OCCAM framework in Section 2, where we describe a range of object-centric masking strategies that vary in their level of abstraction. Section 3 presents our empirical evaluation across Atari environments, focusing on generalization under distribution shifts, shortcut mitigation, and performance relative to pixel and symbolic baselines. After touching on related work, we discuss the limitations and conclude.

## 2 OBJECT-CENTRIC ATTENTION VIA MASKING

Parallel to how humans abstract visual scenes: rather than processing every pixel, we intuitively segment and attend to salient objects while ignoring irrelevant detail (Treisman & Gelade, 1980). The OCCAM principle formalizes such abstraction as a controllable design choice for input representation in RL. Rather than learning object-centric features end-to-end or relying on symbolic pipelines, OCCAM uses lightweight masks to filter distracting visual details while retaining task-relevant structure. The idea is that unnecessary input often enables shortcut learning, brittleness, or adversarial exploitation (Ma et al., 2020; Hermann et al., 2024), whereas reducing such distractions can improve robustness (Bertoin et al., 2022; Grooten et al., 2024). In spirit, OCCAM follows *Occam's Razor*, favoring simplicity over excess, and echoes *Grice's Maxims* (Grice, 1975), emphasizing inputs that are informative and relevant.

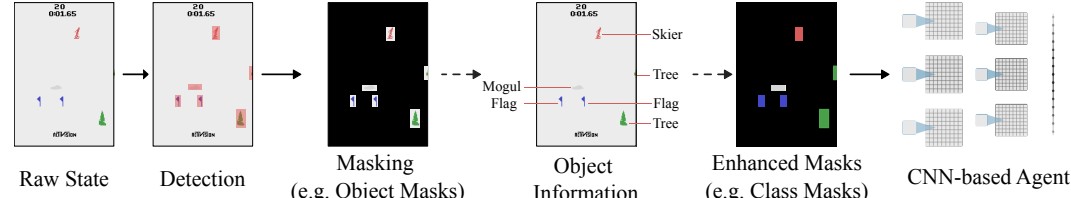

Raw State    Detection    Masking (e.g. Object Masks)    Object Information    Enhanced Masks (e.g. Class Masks)    CNN-based Agent

Figure 1: **Object-Centric Attention via Masking Pipeline.** An object detection method detects moving objects, allowing it to mask out the irrelevant background details. Depending on the information extracted, object features can be used to enrich the masks. This representation is then passed to the CNN-based agent. Here displayed Object Masks and Class Masks, other masks are visualized in Figure 2.

Table 1: **Comparison of OCCAM masking strategies.** Each representation trades off between the amount of visual detail retained, the need for object-class information, and the resulting input dimensionality. For reference, we also include the raw DQN-like pixel input (Mnih et al., 2013) and the symbolic Semantic Vector baseline from OCAtari (Delfosse et al., 2024a). Here, $n$ is the number of objects and $k$ the number of features per object. Plane Masks expand the channel dimension to one per object class, which we refer as $m$.

| Mask Type | Visual Detail Retained | Requires Object Classes | Input Dimensionality |
|---|---|---|---|
| DQN-like (Mnih et al., 2013) | All | No | $4 \times 84 \times 84$ |
| Binary Masks | Position, Size | No | $4 \times 84 \times 84$ |
| Object Masks | Color, Shape, Position, Size | No | $4 \times 84 \times 84$ |
| Class Masks | Object Class, Position, Size | Yes | $4 \times 84 \times 84$ |
| Plane Masks | Object Class, Position, Size | Yes | $4m \times 84 \times 84$ |
| Semantic Vector (Delfosse et al., 2024a) | Object Class, Position | Yes | $2 \times n \times k$ |

To generate structured inputs, OCCAM assumes access to coarse object information (e.g., bounding boxes or segmentation masks) sufficient to define object-centric views. The specific extractor is not the focus of this work: objects could come from engine annotations, pretrained detectors, or simple heuristics. Our aim is not to develop new perception pipelines, but to study how different abstraction levels affect learning once object information is available. For experiments, we rely on OCAtari (Delfosse et al., 2024a) as a controlled source of annotations, but the principle itself is extractor-agnostic. In our experiments, an oracle provides ground-truth object annotations, allowing us to isolate the effect of different representation choices for scientific reasons, while holding detection quality fixed. Our goal is to study how the abstraction level affects learning. For this, we remove influences from different object extractors. In Appendix E we investigate how OCCAM behaves when this assumption is relaxed through synthetic detector noise and make a small extension to learned detectors, such as YOLO (Redmon et al., 2016) and RT-DETR (Lv et al., 2023). The complete OCCAM pipeline is illustrated in Figure 1. OCCAM thus positions abstraction as a design axis for deep RL. It shows how simple masking can control the amount of object-centric information provided to CNN-based agents, making object-centric RL practical and accessible, while clarifying when minimal signals suffice and when a richer structure is needed for robust generalization.

## 2.1 OCCAM'S ABSTRACTION LEVELS

Building on this principle, OCCAM defines a spectrum of masking-based input representations that vary in how much object detail they preserve. While masking and filtering have been explored in RL—for example, unsupervised keypoint discovery that yields sparse object-like landmarks (Kulkarni et al., 2019; Wang et al., 2020), or saliency- and reward-guided masks that suppress irrelevant pixels (Bertoin et al., 2022; Grooten et al., 2024)—these approaches either provide weak, category-agnostic signals or are tied to specific pipelines and their constraints. OCCAM evaluates representative masks through the lens of object-centric reinforcement learning, treating abstraction as a primary design decision. All variants preserve the spatial resolution of pixel observations and remain compatible with standard CNN backbones; Object/Binary/Class Masks plug in without architectural changes, whereas *Plane Masks* preserve resolution but expand the channel dimension,

which we accommodate with a trivial first-layer adaptation (see Table 1). This framing allows us to isolate how representational choices alone shape learning and generalization dynamics, and positions masking as a practical lever for robust and accessible OCRL. We evaluate four representative masking strategies under the OCCAM framework, illustrated in Figure 2:

**Object Masks:** Preserve the full shape, color, and spatial extent of detected objects while masking all background pixels. This strategy resembles classical foreground segmentation methods in computer vision (Shotton et al., 2006; Brox & Malik, 2010) and has been adopted in RL to highlight task-relevant entities by removing background clutter (Anand et al., 2019; Dittadi et al., 2021). While it maintains visual richness and structural detail, it may also retain task-irrelevant features.

**Binary Masks:** Reduce input to a binary occupancy map: pixels within object bounding boxes are set to 1, all others to 0. This resembles foreground–background binarization in computer vision (Otsu, 1979), and has recently been explored in RL to emphasize object locations while discarding appearance cues—for example, in discovering generalized value functions from object features (Nath et al., 2023), or plug-and-play object-centric modules for visual control (Jiang et al., 2023; Shi et al., 2024b). While this representation aggressively removes spurious detail such as color or texture, it can also discard semantic information, making it hard to distinguish between different object types or functions, e.g., a friendly ghost and an enemy ghost.

**Class Masks:** Encode object category information by assigning a distinct value to each object class. This resembles classical semantic segmentation in computer vision (He et al., 2017), and has been adopted in object-centric RL where categorical abstractions support reasoning and generalization (Locatello et al., 2020; Delfosse et al., 2024b; Kohler et al., 2024). Visual detail is still removed entirely, leaving only categorical cues, making this representation semantically richer than Binary Masks but also task-specific, since it requires access to object class labels.

**Plane Masks:** Inspired by structured multi-channel inputs in domains like chess and Go (Browne, 2014; Silver et al., 2016; Czech et al., 2024), Plane Masks allocate one binary channel per object class, with each channel marking the spatial presence of a single object type. Similar disentangled formats have been explored in RL to improve generalization and interpretability (Davidson & Lake, 2020). While this representation makes object structure explicit, its dimensionality scales with the number of classes, and it departs from the plug-and-play format of more compact abstractions.

Each masking strategy marks a point along the abstraction spectrum, embedding distinct inductive biases. Plane-like representations promote spatial disentanglement but increase input complexity and rely on predefined classes. Binary Masks offer simplicity, yet risk discarding critical semantic cues. OCCAM reframes input design in reinforcement learning as an abstraction problem. Rather than prescribing a single representation, it evaluates systematically how perceptual structure impacts generalization. By decoupling perception from architecture and grounding inputs in object-centric priors, OCCAM highlights, not as a fixed method, but as a design perspective: Abstraction helps when aligned with the task's underlying structure. While the importance of representation has long been acknowledged, OCCAM brings the benefits of OCRL into CNN-based agents without requiring bespoke architectures or symbolic pipelines.

## 3 EMPIRICAL EVALUATION

We evaluate OCCAM to test whether lightweight, mask-based abstractions can make object-centric reinforcement learning practical and effective within standard CNN-based agents. Our experiments focus on how different levels of abstraction influence robustness, shortcut avoidance, and baseline comparability. Concretely, we address the following research questions:

**(Q1)** Do OCCAM abstractions improve robustness when agents are evaluated under visual perturbations beyond their training distribution?

**(Q2)** Can OCCAM reduce failure modes such as spurious correlation in *Pong*, where pixel agents overfit to opponent paddle position?

**(Q3)** How well can simple abstractions match or even exceed the performance of strong pixel-based and symbolic baselines while remaining easy to integrate with CNNs?

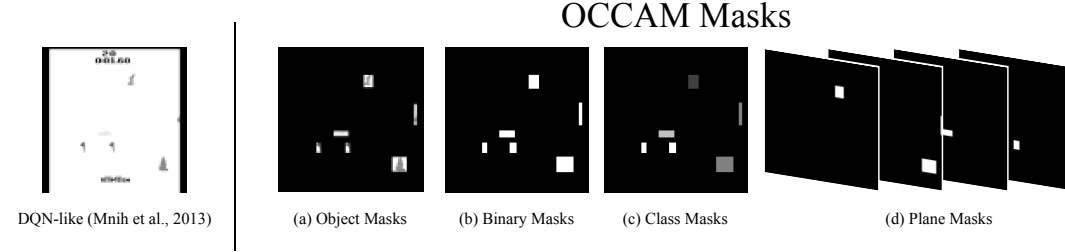

DQN-like (Mnih et al., 2013)      (a) Object Masks      (b) Binary Masks      (c) Class Masks      (d) Plane Masks

Figure 2: **The representations of OCCAM.** Left: DQN-like pixel input after gray-scaling and resizing. Right: the four object-centric masks used in OCCAM: (a) Object Masks preserve only objects, setting background information to 0, (b) Binary Masks whiten bounding boxes, (c) Class Masks assign categories (values) to object boxes, and (d) Plane Masks separate masks into one channel per class. More examples are shown in Appendix F.

EXPERIMENTAL SETUP

We evaluate OCCAM on the HackAtari benchmark (Delfosse et al., 2025), which introduces controlled *visual* and *logical* perturbations (e.g., texture/color shifts, geometry changes, altered opponent policies). Agents are trained *only* on the clean environments and evaluated zero-shot on both clean and perturbed variants (no fine-tuning or test-time augmentation), so reported performance reflects *zero-shot generalization*. We compare against two baseline families: (i) *pixel-based* PPO (Schulman et al., 2017), DQN (Mnih et al., 2013), MDQN (Vieillard et al., 2020), and Rainbow (Hessel et al., 2018); and (ii) *symbolic* OCAtari agents (Delfosse et al., 2024a) that consume compact object features.

While PPO naturally benefits from simplified inputs, we also evaluate a second backend (Rainbow) to show that the effects of OCCAM's abstractions are not tied to a specific architecture. OCCAM representations are constructed via OCAtari. This acts as a controlled oracle for studying representation effects independently of detection errors. Masked and pixel agents share the same network architecture and hyperparameters. Planes, as an exception, has to adapt the first layer size to align with the larger representation. We do this *without additional tuning*. Plane networks are around 10-15% larger in parameter count due to this change. We use standard Atari preprocessing (sticky actions, frameskip), following roughly Machado et al. (2018), train for 40M frames with 3 seeds, and evaluate with 10 episodes per game and seed. Perturbed variants follow HackAtari protocols. As primary metrics, we report the *Interquartile Mean (IQM)* of episodic return with *95% stratified bootstrap* confidence intervals, following Agarwal et al. (2021). IQM averages the central 50% of the return distribution to provide a robust central tendency. Secondly, we are using *Human-Normalized Score (HNS)*, which rescales an agent's performance so that random play corresponds to 0 and average human play to 1, enabling game-agnostic comparisons (values above 1 indicate performance better than humans, below 0 worse-than-random). Human performances are taken from Badia et al. (2020). Per-game tables and extended results appear in Appendix G, while an extended version of the setup can be found in Appendix B.

## 3.1 GENERALIZATION UNDER ENVIRONMENTAL VARIATIONS

We begin our empirical analysis by addressing **Q1**, which concerns the performance of object-centric abstractions in out-of-distribution settings. Specifically, we evaluate whether the benefits of OCCAM persist when agents are exposed to previously unseen variations in visual appearance or environment dynamics. To this end, we leverage the HackAtari benchmark (Delfosse et al., 2025), which introduces controlled perturbations across a set of Atari environments. These modifications fall into two broad categories: (a) **visual changes**, such as recoloring or background replacement, which preserve the underlying game mechanics; and (b) **gameplay changes**, which alter transition dynamics through modifications to object behavior and positioning, enemy logic, or movement timing. We assess generalization by comparing agent performance in perturbed environments, as reported in Figure 3. This comparison quantifies the robustness of each input representation under visual and gameplay distribution shifts, providing a systematic evaluation of how OCCAM's abstractions support out-of-distribution generalization.

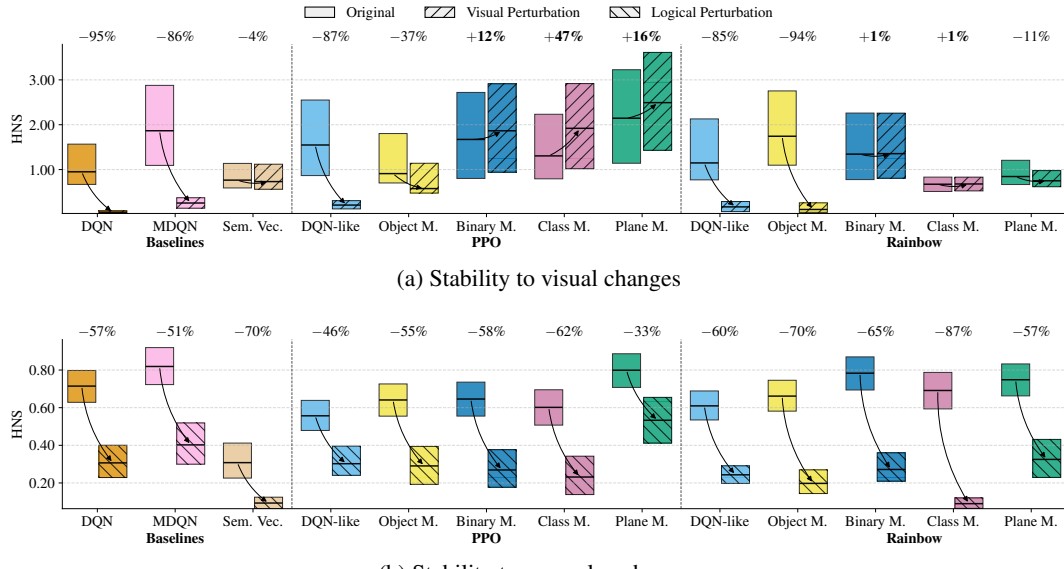

(a) Stability to visual changes

(b) Stability to gameplay changes

Figure 3: **OCCAM improves deep RL when facing visual perturbations.** This figure summarizes stability across agents and environments. OCCAM masks (see Section 2.1), except Object Masks, consistently outperform pixel and symbolic baselines under visual shifts (a). However, this advantage cannot be found under gameplay modifications (b). Results show IQM over the HNS across 3 seeds and 10 evaluation episodes per game. Extended results are in Appendix G.

**Visual Generalization.** OCCAM-based agents, particularly those using Binary and Plane Masks, consistently outperform pixel and symbolic baselines under visual perturbations. By filtering out textures, colors, and other task-irrelevant details, these representations encourage policies that rely on stable spatial structure, such as object positions and motion trajectories. This improves transferability to altered visual conditions and reduces overfitting to superficial features. These findings support OCCAM's core hypothesis: Abstraction, when aligned with object-centric structure, enhances robustness by suppressing spurious correlations and focusing on relevant details.

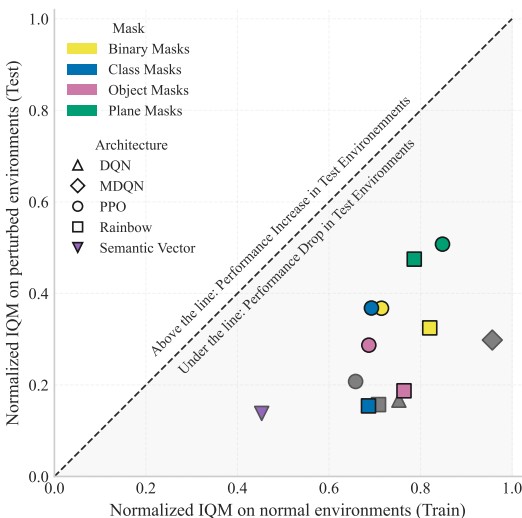

Figure 4: **Trade-off between clean and perturbed performance.** Each point corresponds to an agent; colors denote masking strategies and marker shapes denote base architectures. Points closer to the diagonal indicate stable transfer. OCCAM variants cluster nearer to the diagonal compared to pixel or symbolic baselines.

**Gameplay Generalization.** While OCCAM excels at handling visual variation, its advantages are more limited in the presence of gameplay perturbations that alter environment dynamics. In games like *Frostbite*, where object positions are modified, performance degrades across all representations, including OCCAM. This outcome is expected: when the environment's transition dynamics change, input-level abstraction alone is not sufficient. Addressing these types of shifts likely requires extending OCCAM with mechanisms for temporal reasoning, online adaptation, or structure-aware modeling. This also relates directly to **Q2**, where we explore whether abstraction reduces shortcut-driven behaviors or overfitting to unstable patterns in both perception and dynamics.

Table 2: **Having the correct inductive bias can reduce misalignment.** Performance comparison of different representations in the Pong environment with and without a lazy or hidden enemy. The table shows results for both algorithms: PPO (white) and Rainbow (gray). **Bold** values highlight performance retention for perturbed variants. Values present the IQM together with the confidence interval (for more details see Appendix C). Extended results in Appendix G.

| | Game | Object Masks | Binary Masks | Class Masks | Plane Masks |
|---|---|---|---|---|---|
| PPO | Pong | 18.19 [18,19] | 19.13 [18,20] | 19.44 [19,20] | 18.81 [18,19] |
| | *Lazy Enemy* | 13.81 [12,15] | 10.88 [4,16] | −1.0 [−6,4] | **17.56** [**17,18**] |
| | *Hidden Enemy* | **16.94** [**16,18**] | **19.00** [**18,20**] | **19.19** [**19,20**] | **18.94** [**18,20**] |
| Rainbow | Pong | 20.56 [20,21] | 19.69 [19,20] | 20.67 [20,21] | 20.0 [20,20] |
| | *Lazy Enemy* | −9.12 [−15,−1] | −7.13 [−14,2] | −18.88 [−20,−17] | **16.0** [**15,17**] |
| | *Hidden Enemy* | **19.25** [**18,20**] | **18.12** [**17,19**] | **18.06** [**16,20**] | **19.06** [**19,19**] |

**Interpretation.** As shown in Figure 4, OCCAM variants, particularly Binary and Plane Masks, cluster closer to or above the diagonal, indicating more stable transfer to perturbed environments. Overall, these results demonstrate that OCCAM effectively mitigates perceptual overfitting by reducing reliance on visually spurious cues. Across multiple visually perturbed games, OCCAM-based agents exhibit more stable and reliable behavior than both pixel-based and symbolic baselines.

## 3.2 EXAMINING OCCAM'S EFFECT ON SHORTCUT LEARNING

To evaluate whether OCCAM reduces reliance on spurious correlations, we focus on the *Pong* environment, a well-known case of shortcut learning in deep RL (Delfosse et al., 2024b; 2025). Agents trained on raw pixels often learn to base their policy on the opponent paddle's position, which strongly correlates with the ball trajectory during training. We compare agent performance in standard Pong and two perturbed variants: (1) Lazy Enemy, a logical perturbation, where the opponent paddle stops moving when the ball is returning, breaking the correlation between enemy paddle and ball position. Secondly, (2) Hidden Enemy, a visual perturbation where the opponent paddle is invisible and not registered as object for masking purposes. Both perturbations disrupt shortcut cues while preserving core task dynamics. As shown in Table 2, both pixel-based (PPO, Rainbow) and symbolic (Semantic Vector) agents experience catastrophic failure in both perturbations, while most OCCAM-based agents, particularly the PPO-based ones, retain similar performance. This result aligns with the observation of Hermann et al. (2024), who argue that agents tend to latch onto easily decodable but causally irrelevant features. In the case of Pong, OCCAM mitigates this tendency by masking background pixels and emphasizing localized, object-centric structure. Importantly, it does so without relying on symbolic reasoning, suggesting that rich symbolic object representations may not be strictly necessary for avoiding certain visual shortcuts, contrary to claims by Delfosse et al. (2024b). Lightweight abstractions like OCCAM can achieve similar benefits through structural bias alone.

Surprisingly, the spurious correlation between the ball and opponent position remains present in our masked input representations; yet, OCCAM-based agents consistently learn to ignore or deal with it. We hypothesize that the reason is an inductive bias in early convolutional layers. Masks create sharp spatial gradients between object and background regions, amplifying signals from task-relevant entities while reducing attention to static correlations. This could guide the network toward policies based on dynamics rather than learning to exploit the opponent paddle as a shortcut.

**Beyond Pong: Limits of Shortcut Mitigation.** However, abstraction alone does not fully prevent shortcut-driven behavior, particularly when shortcuts emerge from implicit game dynamics rather than visual features. In *Freeway*, for example, agents often learn to cross safely by exploiting the pattern within the cars, and fail catastrophically when vehicles stop moving. In *BankHeist*, agents learn to ignore the maze structure entirely, instead learning brittle policies that rely purely on quickly switching the level, this is called *reward hacking*. In both cases, OCCAM did not provide any advantage compared to our baselines. Regarding **Q2**, our findings suggest that OCCAM effectively reduces reliance on visual shortcuts, like Pong, it remains vulnerable to structural or logical ones. To summarize, abstraction improves generalization in some regimes, but extending robustness to dynamic or causal confounding remains an open challenge.

Table 3: **Over-abstraction hurts performance compared to baseline.** Performance on *Riverraid* (top row) and its *Linear River* variant (bottom row) across input representations. Each cell shows PPO (white) and Rainbow (grey) scores as IQM + CI. While OCCAM-based agents perform competitively in the original environment, removing spatial layout cues significantly impairs generalization in the modified variant.

| | Game (Variation) | DQN-like | Object Masks | Binary Masks | Class Masks | Plane Masks |
|---|---|---|---|---|---|---|
| PPO | Riverraid | 7668 [7278,8050] | 7784 [7659,7917] | 7908 [7741,8075] | 7714 [7522,7913] | 8020 [7826,8216] |
| | *Linear River* | 6932 [5775,7721] | 2878 [2750,3131] | 2698 [2690,2712] | 2885 [2724,3146] | 2916 [2748,3166] |
| Rainbow | Riverraid | 3009 [2205,4162] | 2482 [2482,2482] | 3383 [2842,4066] | 2888 [2521,3132] | 3043 [2715,3390] |
| | *Linear River* | 5018 [3753,6349] | 2508 [2508,2508] | 3269 [3131,3378] | 2998 [2558,3153] | 3282 [3192,3375] |

### 3.3 Effect of Abstraction on Training-Environment Performance

While earlier results focused on generalization and shortcut mitigation, a critical question remains: How does abstraction influence performance in the training environments? To evaluate this, we train and test all agents exclusively on the original game setting and compare their performance. Baseline agents receive stacked grayscale frames (Mnih et al., 2013), while symbolic agents use compact object-attribute vectors. Each OCCAM variant is evaluated with both PPO and Rainbow backends. Complete per-mask results are reported in Appendix G, along with paired Wilcoxon tests and bootstrap confidence intervals to assess statistical significance. We attempted to include MaDi (Grooten et al., 2024) as a baseline, to compare learned masking against object-centric masking. Under our training budget and a faithful reimplementation (using PPO instead of SAC), MaDi's performance on Atari was substantially below our other baselines, preventing a fair comparison. We omit MaDi from the main plots and report preliminary results and full configurations in Appendix G.

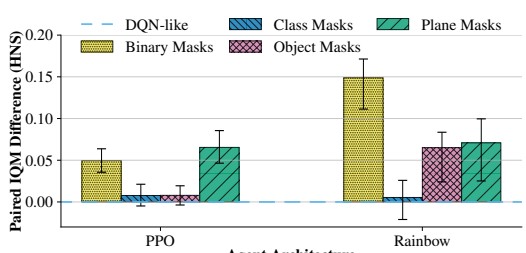

Figure 5: **OCCAM-based representations match or surpass pixel inputs.** Paired IQM differences between each OCCAM variant and its respective baseline across all environments (PPO on the left, Rainbow on the right). Positive values indicate performance improvements over the baseline. Error bars represent 95% confidence intervals estimated via stratified bootstrap. Extended results and further significance tests are provided in Appendix G.

As shown in Figure 5, OCCAM-based agents consistently match or outperform pixel-based baselines, despite relying on sparser, abstracted inputs. PPO tends to benefit more from OCCAM representations than Rainbow. We hypothesize that this reflects differences in learning dynamics: value-based methods such as Rainbow may be more sensitive to the loss of fine-grained pixel information required for accurate value estimation, whereas policy-gradient methods can more readily exploit simplified state spaces. This discrepancy across architectures warrants further investigation, as it may reflect deeper interactions between abstraction, optimization dynamics, and representation capacity.

**Balancing Simplicity and Sufficiency.** While OCCAM reduces input complexity and encourages generalization, abstraction also introduces trade-offs in terms of task-relevant information. In spatially structured environments like *Riverraid*, masking can remove crucial background features such as the river layout, which defines the navigable space. Without this global structure, agents may resort to memorized paths or reactive heuristics that lack flexibility. As shown in Table 3, PPO and Rainbow agents perform competitively in the original environment. However, across the *Linear River* variant, all OCCAM-based abstractions underperform relative to the pixel baseline. Differences are supported by paired IQM confidence intervals and Wilcoxon tests (Appendix G). OCCAM improves robustness when it suppresses spurious visual information, but can harm performance when it removes structure that encodes task-critical constraints. This emphasizes our core message: *Abstraction must be matched to task structure*.

In relation to **Q3**, our results show that OCCAM can match (or even exceed) the performance of standard baselines, despite operating on significantly reduced inputs. While performance varies by environment, these findings demonstrate that well-chosen abstractions can preserve task-relevant structure without sacrificing effectiveness. OCCAM thus emerges as a practical and general-purpose alternative to pixel-based inputs.

Overall, these results also provide guidance on when to favor different levels of abstraction. In most settings, *Planes* emerge as the most effective choice: by disentangling object categories into separate channels, they deliver strong robustness and generalization benefits, and should be preferred whenever the additional training cost is acceptable. For simpler robustness needs, *Binary Masks* offer a computationally lighter alternative while still suppressing many spurious cues. At the same time, our results caution against discarding pixels entirely—raw inputs can retain valuable information, such as navigable layouts in *Riverraid*, that overly aggressive abstraction may erase. In line with the OCCAM principle, the takeaway is not that more abstraction is always better, but that *seeing less, but seeing right* depends on aligning the chosen representation with the task demands.

## 4 RELATED WORK

**Abstraction.** Attending to task-relevant structure while discarding irrelevant detail is central to both human cognition and generalization in RL. Humans reason over object-centric and causal representations (Baars, 1993; 2002; Bengio, 2019; Ho et al., 2019; 2022; Goyal & Bengio, 2022), including in artificial domains like games (Allen et al., 2024). Motivated by this, RL work has explored inductive biases via constrained attention and simplified input design (Zhao et al., 2021; Alver & Precup, 2024). In parallel, theory formalizes state abstraction, showing that aggregating states by reward/policy equivalence can preserve near-optimal behavior (Abel et al., 2016; Abel, 2020).

**Object-centric and symbolic representations.** Object-centric methods improve generalization by encoding scenes through entities and relations (Yoon et al., 2023). Unsupervised discovery approaches, such as SPACE and Slot Attention, learn object-like slots (Lin et al., 2020; Locatello et al., 2020). Meanwhile, unsupervised keypoint extraction provides compact, label-free, object-aware states (Kulkarni et al., 2019). Beyond perception, symbolic/relational policies (trees, logic programs) offer interpretability and structure (Bastani et al., 2018; Delfosse et al., 2024b; Marton et al., 2024; Shindo et al., 2025; Luo et al., 2024). Recent *category-aware* and *entity-centric* RL explicitly condition on object categories or entities to improve manipulation and control (Yi et al., 2022; Haramati et al., 2024). Recent work also investigates policy architectures and world models explicitly tailored to object-centric inputs, often using slot-based encoders or set-based architectures (e.g., Transformers) to process entities and achieve strong compositional generalization and manipulation performance (Zadaianchuk et al., 2020; Haramati et al., 2024; Ferraro et al., 2025; Zhang et al., 2025). In contrast, OCCAM maintains classical architectures designed for visual input, demonstrating that even these standard Atari-style agents can benefit from object-centric abstractions without specialized relational or set-based models.

**Masking, attention, and augmentation in visual RL.** Soft attention mechanisms guide policies toward salient regions but add parameters and training instability (Mott et al., 2019; Manchin et al., 2019). Saliency-driven approaches, such as SGQN, weigh inputs using gradient-based saliency to improve generalization (Bertoin et al., 2022). Reward-driven masking learns a lightweight Masker to suppress distractions directly from returns, showing strong transfer in simulation/robotics (Grooten et al., 2024). Orthogonally, data augmentation injects invariances at the pixel level (RAD, DrQ) and is effective with minimal overhead (Laskin et al., 2020; Yarats et al., 2021). Spectral masking targets frequency-domain nuisances (Huang et al., 2022c), while self-supervised masking objectives (e.g., Mask-based Latent Reconstruction; MIMEx for intrinsic rewards) improve representations and exploration without labels (Yu et al., 2022; Lin & Jabri, 2023). Additionally have recent works leverage segmentation masks or object-aware abstractions to improve control and transfer: Gmelin et al. (2023) disentangle agent and environment representations; Lepert et al. (2024) and Hutson et al. (2024) use masks for cross-embodiment transfer and distraction-robust model-based RL; Wang et al. (2023) and Shi et al. (2024a) use foundation models such as SAM for generalizable visual RL and manipulation, showing that segmentation can indeed help. These works focus on how to obtain or learn the masks, how to make SAM usable for RL, whereas OCCAM is complementary: it assumes object information and instead investigates how these should be represented.

**Generalization, robustness, and input design.** Pixel-based agents often overfit to superficial cues and degrade under visual or dynamic shifts (Farebrother et al., 2018; Delfosse et al., 2025; Hermann et al., 2024). Domain randomization (Tobin et al., 2017), augmentations (Yarats et al., 2021; Laskin et al., 2020), and invariant objectives (Zhang et al., 2021) mitigate but do not explicitly encode object structure; saliency/causal methods (Bertoin et al., 2022) improve interpretability but depend on estimator stability.

## 5 CONCLUSION

This paper introduced OCCAM, a principle that makes object-centric reinforcement learning practical within standard CNN-based agents. By applying lightweight, mask-based abstractions, OCCAM filters away distracting visual details while preserving task-relevant structure, allowing CNN pipelines to benefit from object-centric biases without requiring specialized architectures or symbolic pipelines. Across controlled Atari experiments, OCCAM-based agents consistently match or outperform pixel and symbolic baselines, while improving robustness to visual perturbations and mitigating shortcut failures such as *Pong*. From Binary Masks to per-class Plane Masks, we show that even simple, hand-crafted abstractions can make object semantics explicit. These findings demonstrate that modest representational biases at the perception level can substantially affect generalization and provide a practical design axis for adapting abstraction to task demands. At the same time, our results reveal the limits of over-simplification: tasks differ in the structure they require, and sensitivity to dynamics underscores the need for *adaptive abstraction*, where input granularity adjusts to context and complements other robustness tools. Reliable and aligned RL will not come from scaling models alone, but from making representation a central design choice.

Our work isolates the role of *input abstraction* in Atari as a controlled first step. While this setting enables clean comparisons, it also imposes constraints. First, our evaluation is limited to Atari/HackAtari, which offers uniquely controllable perturbations but does not capture the diversity of real-world noise or complex dynamics. Extending OCCAM to other benchmarks, such as Procgen (Cobbe et al., 2020), Crafter (Hafner, 2022), or robotics (Jiang et al., 2023; Shi et al., 2024b), would strengthen it. Second, OCCAM addresses visual robustness and perceptual shortcuts; our results on gameplay/dynamics perturbations (e.g., Freeway, BankHeist) show that input abstraction alone is insufficient for structural or causal shifts, which require complementary approaches, e.g., model-based or causal methods. Further, our masks rely on privileged engine annotations. Although this provides a controlled testbed, it sidesteps the challenges of imperfect perception. To begin addressing this, we provide a small synthetic study of noisy object extraction, as well as a short ablation about the usage of YOLO or RT-DETR in Appendix E.

Beyond these limitations, we identify three promising directions. (i) *Compfarative evaluation:* systematically benchmark OCCAM against other robustness strategies, including augmentation-based methods (RAD, DrQ (Laskin et al., 2020; Yarats et al., 2021)), saliency-driven filtering (SGQN (Bertoin et al., 2022)), and learned masking (MaDi (Grooten et al., 2024)), to position object-centric abstraction in the broader landscape. Learned masking in particular may offer a path to avoid over-simplification. (ii) *Robustness to imperfect perception:* explicitly study how OCCAM degrades under missing detections, jittered positions, or class confusion, as started in Appendix E. (iii) *Adaptive abstraction:* develop mechanisms that modulate the level of masking online, e.g., adjusting between raw pixels, binary masks, class-wise channels, or hybrid representations based on task phase, uncertainty, or environment complexity. Our preliminary experiments in Appendix G demonstrate that naively combining masked and pixel-based inputs can even further improve agents, while retaining most of the robustness gained from the mask. This needs more insight, but it is clearly an interesting direction. Further future work could explore meta-learning or curriculum-based approaches that balance representational simplicity with task sufficiency.

## ETHICS STATEMENT

This work uses only publicly available Atari environments and the HackAtari benchmark, which introduces controlled perturbations for testing robustness. While some of these games contain abstracted representations of combat or "shooting," the use of such environments in this work serves solely as a standardized benchmark for evaluating reinforcement learning algorithms. No human

subjects, sensitive data, or personally identifiable information were involved. We acknowledge that object extraction pipelines may introduce biases or failure modes, and care must be taken to evaluate fairness, robustness, and potential downstream risks. All authors have read and adhere to the ICLR Code of Ethics.

## REPRODUCIBILITY STATEMENT

We have made every effort to ensure reproducibility. The main paper specifies the OCCAM framework, masking strategies, evaluation metrics, and baselines (see Section 3). Appendix B details hyperparameters and training procedures, with a more detailed description of the metrics in Appendix C. Extended per-game results and statistical testing appear in Appendix G. We rely on standard open-source toolkits (Gymnasium, OCAtari, HackAtari, CleanRL) and provide pseudocode and usage examples for the object extraction in Appendix D. An anonymized code release will accompany the submission to support full reproducibility, with some models available for testing. They will also be released after proceedings.

**Code** https://anonymous.4open.science/r/occam-78E0

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

## A    USAGE OF LLMs

We acknowledge the use of tools such as ChatGPT, DeepL Write, and Grammarly to assist with writing, grammar correction, and language polishing during the preparation of this manuscript.

## B    EXPERIMENTAL SETUP

To evaluate the effectiveness of object-centric abstraction in RL, we conduct a series of controlled experiments using the Atari Learning Environment (ALE). We benchmark our approach against conventional pixel-based RL methods, including Deep Q-Networks (DQN) (Mnih et al., 2013), Proximal Policy Optimization (PPO) (Schulman et al., 2017), Rainbow Hessel et al. (2018) and MDQN (Vieillard et al., 2020), as well as OCAtari (Semantic Vector) (Delfosse et al., 2024a) as another object-centric representation with focus on symbolic representation. The DQN and MDQN baseline models were taken from Gogianu et al. (2022), while the PPO, Rainbow, and Semantic Vector agents were trained by ourselves. To test robustness, we decided to test the trained agents not only in their original environment but also in their perturbations. These perturbations, derived from HackAtari (Delfosse et al., 2025), include visual alterations (e.g., color changes, object recoloring), structural modifications (e.g., object displacement, swapped game elements), and gameplay variations (e.g., altered agent dynamics or enemy behavior). The metrics used are the human normalized score (HNS), over the aggregated game scores, using the interquartile mean (IQM). Calculations are done with rliable (Agarwal et al., 2021). The metrics are described in Appendix C.

### CODE

All source code, experiment configurations, and processed evaluation data used in this work are available at https://anonymous.4open.science/r/occam-78E0. The repository includes implementations of the proposed OCCAM framework, training scripts for Atari environments, to facilitate reproducibility.

### ENVIRONMENT SELECTION

We evaluate our framework across a diverse set of 11 Atari games selected to balance reactive and strategic decision-making tasks (Boxing vs. Freeway), having environments where the background information or object features are crucial for the game (MsPacman), as well as some classics, such as Pong and Breakout. This selection allows us to assess the adaptability of different representations across a spectrum of task complexities. To list all environments again, we used Amidar, BankHeist, Bowling, Boxing, Breakout, Freeway, Frostbite, MsPacman, Pong, Riverraid, and SpaceInvaders.

### TRAINING PPO

PPO is a policy gradient algorithm widely used in deep RL that optimizes a clipped surrogate objective to balance exploration and stability (Schulman et al., 2017). Unlike value-based approaches, such as DQN, PPO directly learns an optimal policy distribution and is known for its sample efficiency and stable convergence properties. It is one of the most common architectures used in RL and, as such, is a good baseline and base for our experimental section. We trained six PPO agents for the experiments using varying input representations. This allowed us to isolate the trade-offs between abstraction strength and spatial reasoning capacity and evaluate our visual reasoning with object-centric attention. All agents are trained on the unmodified versions of these environments for 40 million frames using PPO, with hyperparameters adapted from Huang et al. (2022b) and listed in Table 5. For PPO, we adhere to standardized implementation guidelines (Huang et al., 2022a) to ensure comparability and use the aforementioned CleanRL framework (Huang et al., 2022b) for training. The object-centric representations, derived using OCAtari, selectively preserve task-relevant features while eliminating extraneous background information. Importantly, no fine-tuning is performed in perturbed environments, ensuring that generalization performance reflects the robustness of the learned representations rather than additional adaptation. Performance is evaluated using average episodic rewards across 10 games per seed, with three fixed seeds per experiment. All experiments are conducted on NVIDIA V100 GPUs, with training times averaging around 2–6 hours per agent per seed for the DQN-like, Object Masks, Binary Masks, and Class Masks. Training

the Semantic Vector representation took less than 2h (avg. 1h 38min), while for Plane Masks, the environment highly influenced the training time and took 6 to 10h per game per seed. Overall, it took roughly 900 GPU hours to train all PPO agents.

### Training Rainbow

Rainbow is a value-based deep RL algorithm that integrates several improvements over the original DQN architecture, including prioritized replay, n-step returns, distributional value learning, and dueling networks (Hessel et al., 2018). As a strong and widely adopted baseline in Atari benchmarks, Rainbow serves as a useful counterpart to PPO in evaluating the effectiveness of OCCAM-based input abstractions. We trained five Rainbow agents (Rainbow, Object Masks, Binary Masks, Class Masks, Plane Masks) using the same set of input representations as in the PPO experiments. This setup allows us to assess whether the benefits of object-centric abstraction generalize across different learning paradigms, namely, value-based and policy-based methods. All Rainbow agents were trained for 40 million environment frames using the implementation from the CleanRL framework (Huang et al., 2022b), with hyperparameters adapted to match standard evaluation settings, introduced by Dopamine (Castro et al., 2018) (see Table 6). As with PPO, the object-centric inputs are derived using OCAtari, and no fine-tuning is performed on perturbed environments. We, however, had to adapt the replay buffer size for the Plane Masks from $10^6$ to $10^5$ to increase training stability and due to resource limitations. Performance is evaluated using average episodic rewards over 10 episodes per seed, with three seeds per condition. All training was conducted using NVIDIA V100. Training time per agent per seed ranged from 20 to 30 hours for most representations. The huge difference is mainly based on the fact that we did not use parallel environments, as we did with PPO. The total GPU cost of Rainbow training across all seeds and conditions was approximately 3,000 GPU hours. The complete hardware description is given in Table 7.

### Network Architectures and Size

OCCAM abstractions differ in how they encode object information and how easily they integrate with CNN backbones:

**Masks (Object/Binary/Class).** These preserve the spatial format of pixel inputs with four stacked grayscale frames of size $84 \times 84$. Because the channel count is unchanged, they are parameter- and training-neutral relative to pixel inputs, making them drop-in replacements for standard CNN-based agents.

**Plane Masks.** Inspired by structured inputs in board games and object-centric RL, Plane Masks allocate one binary channel per object class. This preserves spatial layout while disentangling categories, but increases input dimensionality from 4 channels to $4m$ channels (where $m$ is the number of classes). As a result, the first convolutional layer grows in size, increasing overall parameters by less than 10%. While the model size change is modest ($\sim 1.7\text{M} \rightarrow 1.8\text{–}1.9\text{M}$), the larger stacks lead to 2–3 times longer training time in practice.

**Semantic Vectors.** Provided by OCAtari, these are compact object feature vectors (class, position, attributes) extracted per frame. They remove spatial layout entirely and must be processed by MLPs rather than CNNs, sacrificing plug-and-play compatibility with pixel pipelines. While efficient in principle, they are fragile under noise or missing labels. For comparability, we upscaled the MLP to yield a similar parameter budget to the CNN baselines.

**Network architecture.** Unless otherwise stated, all agents use the same convolutional backbone as in PPO/Rainbow Atari baselines. The encoder consists of three convolutional layers (32 filters of $8{\times}8$ stride 4, 64 filters of $4{\times}4$ stride 2, 64 filters of $3{\times}3$ stride 1), followed by a linear projection to a $512$-dimensional feature vector. From this shared representation, the actor head outputs action logits and the critic head outputs a scalar value estimate. Mask-based and pixel agents use the same architecture and hyperparameters without modification; only Plane Masks increase the channel dimension of the first convolutional layer. The specific network sizes can differ slightly due to the different action spaces used in the Atari games. This difference is, however, negligible.

Table 4: **Comparison of abstraction types.** All mask variants preserve spatial format and are CNN-compatible; Semantic Vectors drop layout but are compact. Reported parameter counts are approximate for the PPO backbone used in our experiments.

| Representation | Spatial Layout | Input Dim. | CNN-Compatible | Params (M) |
| --- | --- | --- | --- | --- |
| Object/Binary/Class Masks | Yes | $4 \times 84 \times 84$ | Yes | $\sim 1.7M$ |
| Plane Masks | Yes | $4m \times 84 \times 84$ | Yes (larger first layer) | $\sim 1.8$–$1.9M$ |
| Semantic Vector | No | $4 \times n \times k$ | No (MLP required) | $\sim 1.5$–$2.0M$ |

In summary, Masks and Plane Masks enable controlled abstraction while remaining compatible with CNNs, making them practical for OCRL. Semantic Vectors are more compact but less flexible, as they require specialized architectures and lose spatial structure.

SCALABILITY OF PLANE MASKS

While most masks preserve the same number of input channels as pixel agents (four stacked frames), *Plane Masks* allocate a dedicated channel for each object class. This increases the number of input channels linearly with the class vocabulary size and expands the parameter count of the first convolutional layer. In our PPO backbone, this layer constitutes only a small fraction of the total model, so the overall increase is modest: Object, Binary, and Class agents use roughly 1.7M parameters in total, whereas Plane Masks range from 1.8M to 1.9M depending on the number of classes. Thus, even though Plane Masks multiply the parameter count of the first layer, the total network size changes by less than 10%. The main trade-off is computational: larger input tensors increase per-step cost. In our setup, the observed 2-3 times slowdown is largely a hardware artifact; the bottleneck is CPU throughput and the number of parallel environments that can be maintained when handling higher-dimensional inputs. When we reduce parallelism more conservatively (as in our Rainbow runs), the training-time gap shrinks substantially, to roughly 10–20% between Planes and non-planes Masks. In larger-scale settings, this overhead can be mitigated by grouping semantically similar classes into shared planes (e.g., all projectiles, all enemies) or by restricting Plane Masks to a small set of task-relevant categories, keeping the benefits of disentanglement while controlling input dimensionality.

HYPERPARAMETERS

We detail the hyperparameters used in our training in Tables 5, 6 and 8 to ensure reproducibility and consistency in our experiments. These settings were chosen based on prior literature.

Table 5: Key hyperparameters used for **PPO** training, following (Huang et al., 2022a) and (Huang et al., 2022b).

| Hyperparameter | Value |
| --- | --- |
| Seed | $\{0, 1, 2\}$ |
| Learning Rate ($\alpha$) | $2.5 \times 10^{-4}$ |
| Total Timesteps | $10^7$ |
| Number of Environments | 10 |
| Batch Size ($B$) | 1,280 |
| Minibatch Size ($b$) | 320 |
| Update Epochs | 4 |
| GAE Lambda ($\lambda$) | 0.95 |
| Discount Factor ($\gamma$) | 0.99 |
| Value Function Coefficient ($c_v$) | 0.5 |
| Entropy Coefficient ($c_e$) | 0.01 |
| Clipping Coefficient ($\epsilon$) | 0.1 |
| Clip Value Loss | True |
| Max Gradient Norm ($\|g\|_{max}$) | 0.5 |

Table 6: Key hyperparameters used for **Rainbow** training, following (Castro et al., 2018).

| Hyperparameter | Value |
|---|---|
| Total Environment Steps | $10^7$ |
| Replay Buffer Size | $10^6/10^5$ |
| N-step Returns | 3 |
| Value Distribution Atoms | 51 |
| Value Support Range | $[-10, 10]$ |
| Batch Size | 32 |
| Target Network Update Frequency | 8,000 |
| Optimizer | Adam |
| Learning Rate ($\alpha$) | $6.25 \times 10^{-5}$ |
| $\epsilon$-greedy Start | 1.0 |
| $\epsilon$-greedy End | 0.01 |
| $\epsilon$ Anneal Steps | 1,000,000 |
| Discount Factor ($\gamma$) | 0.99 |
| Min Replay Size | 1,600 |
| Update Frequency | Every 4 steps |
| Max Gradient Norm | 10.0 |
| Noisy Nets | `False` |
| Dueling Architecture | `True` |
| Prioritized Replay | `True` |
| Priority Exponent ($\alpha_{\text{prio}}$) | 0.5 |
| Priority Importance Sampling ($\beta_{\text{prio}}$) | $0.4 \rightarrow 1.0$ |

Table 7: **Hardware configuration** used for all experiments. We use an NVIDIA DGX system equipped with V100 GPUs and follow standardized training pipelines.

| Component | Specification |
|---|---|
| System | NVIDIA DGX v.5.1.0 |
| GPUs | $16 \times$ NVIDIA V100 32GB |
| CPUs | Intel Xeon Platinum 8174 |
| System Memory | 1.58 TB |
| Operating System | Ubuntu 20.04 LTS |
| CUDA Version | 12.4 |
| Python Version | 3.10 |

Table 8: Key hyperparameters regarding the used **environments**. We are using Gymnasium and the ALE, following the best practices by Machado et al. (2018) and the community.

| Hyperparameter | Value |
|---|---|
| ALE version | 0.8.1 |
| Gymnasium version | 0.29.1 |
| Environment version | v5 |
| Frameskip | 4 |
| Buffer Window Size | 4 |
| Observation Mode | RGB |
| Repeat Action Probability | 0.25 |
| Full Action Space | `False` |
| Continuous | `False` |

## C  EVALUATION METRICS

We evaluate agent performance using three complementary metrics: (1) **Human-Normalized Score (HNS)** for absolute performance relative to human and random baselines; (2) **Interquartile Mean (IQM)** to aggregate across multiple seeds and environments in a statistically robust manner; and (3) **95% Confidence Intervals (CI)** via stratified bootstrap to quantify uncertainty. These metrics follow best practices from Agarwal et al. (2021) and are computed using the `rliable` library.

### HUMAN-NORMALIZED SCORE (HNS)

The Human-Normalized Score (HNS) is a standard metric in Atari benchmarks (Mnih et al., 2015; Machado et al., 2018), enabling normalized comparison of agent performance across environments with varying reward scales. Given the average agent score $A$, human score $H$, and random score $R$, The HNS is defined as:

$$\text{HNS} = \frac{A - R}{|H - R|} \tag{1}$$

A value of 1.0 (or 100%) corresponds to human-level performance. Scores above 1.0 indicate super-human performance; values around 0 suggest behavior close to a random agent; and negative values imply sub-random performance. We use the human and random baselines reported by Badia et al. (2020), which were obtained from professional human testers.

### AGGREGATION VIA INTERQUARTILE MEAN (IQM)

Raw HNS values can vary substantially across random seeds, especially in environments with stochastic elements or sparse rewards. To ensure that our reported results reflect typical performance rather than outliers, we aggregate agent scores using the **Interquartile Mean (IQM)**. This metric computes the mean over the middle 50% of the data (i.e., between the 25th and 75th percentiles), reducing sensitivity to extreme values and providing a robust summary across multiple runs.

For all experiments, we report the IQM over 30 evaluation episodes (3 seeds × 10 episodes), stratified per environment and representation. Aggregation across games is done using `rliable` (Agarwal et al., 2021), which computes the IQM of the HNS distribution across all environments.

### UNCERTAINTY ESTIMATION WITH 95% CONFIDENCE INTERVALS

To quantify uncertainty, we report **95% stratified bootstrap confidence intervals** alongside IQM values. Bootstrap resampling avoids strong parametric assumptions and provides a reliable estimate of variability, even when return distributions are skewed or heavy-tailed—common in deep RL. These intervals allow for statistically grounded comparison between methods and help guard against overinterpretation of noisy results.

### TESTING STATISTICAL SIGNIFICANCE

To assess statistical significance and aggregate performance differences between OCCAM variants and baseline agents, we use two complementary techniques: paired IQM confidence intervals and per-environment Wilcoxon signed-rank tests.

**Paired IQM Confidence Intervals.**  We compute the IQM over the paired performance differences (i.e., per-seed, per-environment) between each OCCAM variant and its respective baseline. This yields a single aggregated IQM difference per variant. Confidence intervals are estimated using stratified bootstrap resampling, preserving the environment-level structure of the data. This provides a robust, summary-level view of how each variant shifts performance relative to the baseline.

**Wilcoxon Signed-Rank Tests.**  For each environment, we conduct one-sided Wilcoxon signed-rank tests to determine whether an OCCAM variant significantly outperforms or underperforms the baseline. These tests are non-parametric and account for paired differences between methods (across seeds), making them well-suited for small-sample RL evaluations. We report bidirectional $p$-values

in our tables to distinguish between improvement, degradation, or similarity, using a significance level of $\alpha = 0.05$.

Together, these methods quantify both the directionality and consistency of performance changes, providing strong statistical support for claims about representation quality.

**Summary.** Together, all these metrics provide a reliable and interpretable view of agent performance. HNS allows for direct comparison against human-level behavior, making it easy to contextualize how well an agent performs in absolute terms and gives us the possibility to aggregate over multiple games. IQM ensures that comparisons across seeds and environments are not skewed by outlier runs, highlighting the agent's typical performance rather than best-case results. The inclusion of 95% confidence intervals with additional metrics like Wilcoxon adds statistical rigor, indicating whether observed differences are meaningful or could be due to variability in outcomes.

OBJECT DETECTION METRICS.

To assess the quality of YOLO and RT-DETR in Atari (cf. Appendix G), we report standard object-detection metrics. The F1 score (computed after applying a confidence threshold) captures the trade-off between precision and recall, measuring whether the detector can both identify most relevant objects and avoid spurious predictions. For spatial localization quality, we report mean Average Precision (mAP). A detection counts as correct if its Intersection-over-Union (IoU) with a ground-truth box exceeds a threshold $\alpha$. mAP@50 corresponds to $\alpha = 0.5$, a lenient setting where approximate localization suffices. The stricter COCO-style mAP@50–95 averages AP over IoU thresholds $\alpha \in \{0.50, 0.55, \ldots, 0.95\}$:

$$\text{mAP@50–95} = \frac{1}{10} \sum_{\alpha=0.50}^{0.95} \text{AP}(\alpha). \tag{2}$$

This metric penalizes both missed detections and imprecise bounding boxes. Together, $F_1$, mAP@50, and mAP@50–95 quantify both object-identification accuracy and the fidelity of spatial localization.

## D  OBJECT EXTRACTION WITH OCATARI

To extract structured object representations from Atari environments, we leverage **OCAtari** Delfosse et al. (2024a), a lightweight framework built on top of the ALE. In this work, we exclusively use OCAtari's **RAM Extraction Method (REM)**, which decodes semantically meaningful object properties directly from the emulator's memory. These object states serve as input to our masking module for generating structured, task-agnostic representations.

**Installation.**  OCAtari and its dependencies can be installed using:

```
pip install ocatari
pip install gymnasium[atari]
```

**Object Information.**  OCAtari's REM provides access to object-level states (e.g., category, position, size) for each supported game. These mappings are hand-designed per game and remain consistent across runs, allowing for reliable and interpretable abstraction. A minimal usage example is shown in Listing 1.

Each object is returned as a structured entity with (at least):

- `category` (e.g., "Ball", "Player", "Score"),
- `x,y`: pixel coordinates relative to the upper left corner,
- `w,h`: width and height in pixels,
- `dx,dy`: the difference in x and y between this frame and the last.

Head-up display (HUD) elements (e.g., score counters) are optional and were not included in this work. Most HUD elements would be made irrelevant or unhelpful when putting them into our masking approaches, as such they would add additonal noise not helping the agent in their decision. However, in some games HUD elements can play a major role in the decision-making process, such as Seaquest or BankHeist. In such cases it can be helpful to use HUDs and incorporate them as additonal objects or plane in the representation. To do so, change the OCATari parameter `hud` to True.

**Usage.**  OCAtari can be used like any other GymWrapper. An example can be found in Listing 1.

```python
from ocatari.core import OCAtari

env = OCAtari("Pong", mode="ram")
action_space = env.action_space
obs, info = env.reset()

for _ in range(1000):
    #obs, reward, terminated, truncated, info
    s_tuple = env.step(action_space.sample())
    terminated, truncated = s_tuple[2:4]

    for obj in env.objects:
        print(obj.category, obj.xy, obj.wh)
    if terminated or truncated:
        break
```

Listing 1: Using OCAtari with REM to extract object properties in `Pong`.

**Integration with OCCAM.**  The REM-derived object list serves as the basis for our masking pipeline. We generate structured representations, such as bounding box masks, binary bitboards, or class-encoded planes, by projecting object coordinates back into the pixel space. The masks are stacked across time steps and passed to the RL policy, preserving spatial semantics while cutting task-irrelevant features. Due to its determinism and computational efficiency, REM enables high-throughput training with consistent abstraction.

Table 9: **Binary Masks (PPO) with imperfect object detection in Pong.** We evaluate Binary Masks trained with PPO under varying levels of object detection noise. Each object has a 2%, 5%, or 10% probability of being dropped, and Gaussian noise is added to its position. Pong shows little sensitivity to noise: results remain close to the perfect setting (19 [18,19]) across all settings. **Bold** values indicate the best-performing model in each test case.

| Train Fail Probability | Test Fail Probability | | | |
|---|---|---|---|---|
| | 0% | 2% | 5% | 10% |
| 0% | **19** [**18,19**] | **19** [**19,20**] | **16** [**14,17**] | 17 [16,18] |
| 2% | 17 [15,18] | **19** [**18,19**] | **16** [**14,17**] | **18** [**17,18**] |
| 5% | 15 [14,16] | **19** [**18,20**] | **16** [**15,17**] | 15 [14,17] |
| 10% | 13 [11,15] | 18 [17,18] | 15 [14,16] | 17 [16,18] |

**Resources.** For a complete list of supported environments and advanced usage, visit `https://oc-atari.readthedocs.io`.

# E  ALTERNATIVE OBJECT EXTRACTORS AND IMPERFECT PERCEPTION

In the main paper, we rely on OCAtari's REM as a *perfect extractor oracle* to cleanly isolate the effect of input *representations* without conflating it with perception errors. This lets us ask: given object information, how should it be encoded for CNN-based RL? To probe practicality beyond this idealized setting, Let us report a small synthetic study with noisy masks, evaluate YOLO- and RT-DETR–based detectors on Atari frames. We include these analyses in the appendix because they are exploratory and do not constitute a full detector benchmark (no extensive tuning or architecture search); however, they illustrate how OCCAM begins to behave when object information is imperfect, rather than oracle-level.

## ADDING NOISE TO THE DETECTION

To obtain a first indication of OCCAM's robustness under such conditions, we conduct a small, synthetic ablation using Binary Masks with PPO. Each object is randomly dropped from the mask with probability 2%, 5%, or 10%, and its position and size are perturbed with Gaussian noise.

Results show a contrast across environments. In *Pong*, which involves only a few objects, performance remains close to the perfect-extractor setting across all noise levels, indicating resilience to moderate imperfections (Table 9). In *Breakout*, by contrast, policies trained without noise degrade substantially when tested under noisy observations. Introducing moderate noise during training yields stronger policies for noisy object detectors, but performance does not quite achieve that of training and testing with perfect information (Table 10).

We stress that this is a very limited and highly synthetic study, intended as a first dip into the question of perception noise. While the results suggest that OCCAM can tolerate imperfect detection, they cannot substitute for experiments with real learned detectors.

## EXPERIMENTS WITH YOLO AND RT-DETR

To complement the synthetic-noise study above, we conducted a preliminary evaluation using *learned* object detectors. Our goal is not to build a full perception pipeline, but to assess whether off-the-shelf detectors can localize Atari objects with sufficient fidelity to support OCCAM-style abstraction. We evaluate two state-of-the-art detectors from the Ultralytics suite[1]: **YOLO11/YOLO12 (Tian et al., 2025)** and **RT-DETR (Lv et al., 2023)**. To assess whether these detectors can learn Atari object distributions, we construct a large-scale training dataset using OCAtari's REM annotations as labels. All models are initialized from COCO-pretrained weights

[1]`https://www.ultralytics.com/`, accessed 2025-11-14.

Table 10: **Binary Masks (PPO) with imperfect object detection in Breakout.** We evaluate Binary Masks trained with PPO under the same imperfect object detection setup as in Pong. Each object has a 2%, 5%, or 10% probability of being dropped, and Gaussian noise is added to its position. Agents trained with perfect object detectors (0% train fail probability) struggle under noisy test conditions, whereas agents trained with moderate noise perform better under noisy conditions, but cannot keep up with the perfect conditions. **Bold** values indicate the best-performing model in each test case.

| Train Fail Probability | Test Fail Probability | | | |
|---|---|---|---|---|
| | 0% | 2% | 5% | 10% |
| 0% | **166** [**105**,**227**] | 39 [31,48] | 32 [18,53] | 19 [15,28] |
| 2% | 11 [7,14] | **100** [**87**,**114**] | 63 [45,80] | 67 [56,85] |
| 5% | 8 [6,10] | 71 [64,81] | **65** [**52**,**82**] | **72** [**62**,**87**] |
| 10% | 6 [4,7] | 69 [58,83] | 60 [53,70] | 55 [48,64] |

and fine-tuned for 100 epochs on these annotations. We evaluate performance on a held-out test set, also annotated via REM, to measure generalization to unseen game states.

**Training Setup.** For training our agents, we followed the best practices of ultralytics[2], providing datasets with 12 episodes per game (10 training episodes, 2 validation episodes), resulting in around 20,000 frames in training. We use Ultralytics data augmentation methods and default parameters. While we see the importance to optimize the hyperparameters[3], this is out of scope for this work. We trained object detectors for 4 games (i.e., Pong, Freeway, Amidar, MsPacman) for 100 epochs each.

**Evaluation Setup.** To finally evaluate OCCAM with state-of-the-art object detectors, we use the newly trained YOLO and RT-DETR models to replace the object detection of OCAtari. The OCCAM agent is now evaluated on 10 episodes based on this object information. As an initial study, we evaluate the same OCCAM Plane Masks agent for all detector settings.

At the same time, Atari is a domain where specialized vision pipelines can work very well. Several approaches extract object-like structure from Atari frames (Liu et al., 2019; Locatello et al., 2020; Lin et al., 2020; Delfosse et al., 2023a), and some even achieve near-perfect sprite recovery (Smirnov et al., 2021). In this context, the OCAtari Vision Extractor (VEM) serves as a lightweight, vision-based proxy for an oracle detector: it infers Atari objects directly from RGB frames using handcrafted rules and template-style matching, providing approximate bounding boxes and class labels without relying on the emulator's memory state. IoUs of it are provided by the authors in their paper (Delfosse et al., 2024a). Note, OCAtari's VEM did not run on Amidar out of the box.

**Evaluation Results.** The object detection quality was evaluated over 2 episodes and can be seen in Figure 6. YOLO consistently exhibits high F1 and mAP@50–95 scores (over 90% mAP@50 and 80% mAP@50-95), indicating that it reliably localizes Atari sprites when fine-tuned on them. Table 11 shows that OCCAM can also maintain performance comparable to vision-based detectors if object detection is sufficiently accurate (VEM and Freeway). However, if object detection is incomplete or noisy, the performance can change depending on the objects missing. In Pong, YOLO often had difficulties locating the ball, making it hard for the agent to play. In Amidar, however, missing one enemy did not significantly impact performance. This aligns with our findings from the noise study. Furthermore, our experiment demonstrated that YOLO encountered fewer issues when working with Atari game frames compared to RT-DETR, which may be attributed to its smaller size or architectural differences.

---

[2]https://docs.ultralytics.com/modes/train/, accessed 2025-11-14.
[3]https://docs.ultralytics.com/modes/train/#train-settings, accessed 2025-11-14.

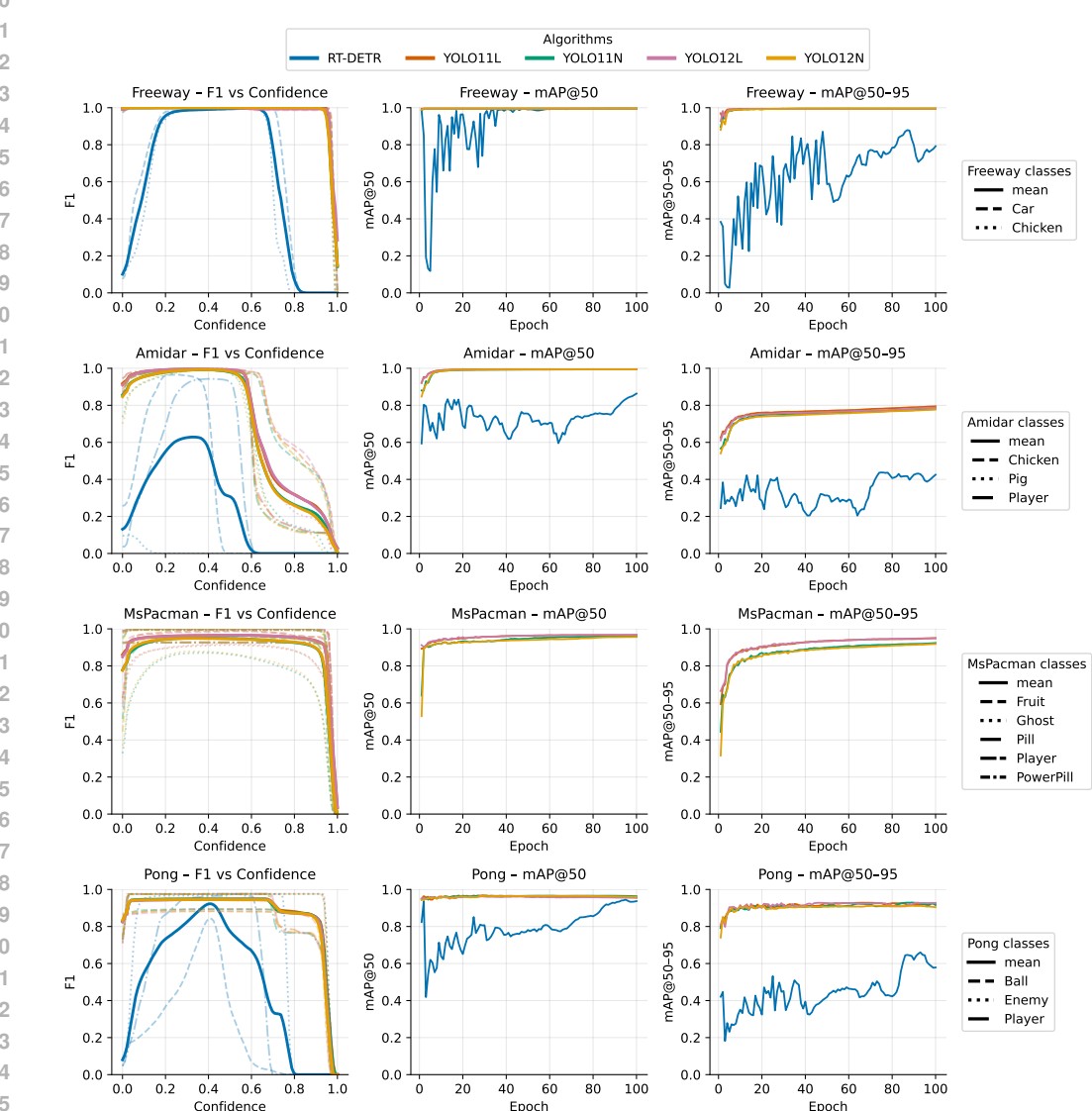

Figure 6: **YOLO and RT-DETR trained on Atari.** Results in training object detectors on 4 Atari games, presenting F1, mAP@50 and mAP@50-95.

Table 11: **Detection Noise can have a strong influence on performance deepening on the objects misdetected.** Performance Evaluation of OCCAM using YOLO and RT-DETR for object detection, evaluating the PPO model with Plane Mask representation. OCCAM REM is displayed as baseline using perfect detection. Setup is the same as and results are comparable to Table 14.

| Detector | Pong | Freeway | Amidar | MsPacman |
|---|---|---|---|---|
| OCAtari REM (Oracle) | 19.4 [18.0,20.4] | 34.0 [33.4,34.0] | 543.4 [531.6,562.2] | 6436.0 [4926.0,7418.0] |
| OCAtari VEM | 19.2 [17.4,20.0] | 34.0 [33.4,34.0] | — | 2126.0 [2062.0,2192.0] |
| YOLO11n | 13.6 [10.2,16.4] | 34.0 [33.8,34.0] | 561.6 [532.6,601.0] | 5620.0 [4868.0,5644.0] |
| YOLO11l | −3.4 [−11.4,5.2] | 34.0 [33.4,34.0] | 593.2 [564.2,594.2] | 2348.0 [2292.0,2572.0] |
| YOLO12n | −3.4 [−11.4,5.2] | 34.0 [34.0,34.0] | 573.4 [500.0,593.2] | 2442.0 [2366.0,2928.0] |
| YOLO12l | −12.0 [−14.4,−10.2] | 34.0 [34.0,34.0] | 632.4 [591.6,642.6] | 2538.0 [2368.0,2952.0] |
| RT-DETR | −7.8 [−14,6,2.0] | 8.8 [7.2,10.0] | 1.0 [0.4,5.8] | — |

## F    INPUT REPRESENTATIONS IN FROSTBITE AND MSPACMAN

Figure 2 displays the input representations used in our experimental section, illustrating their differences and highlighting which elements are retained or abstracted. With Figures 7 and 8, this section presents two additional examples, applying the exact representations to two more games, i.e., Frostbite and MsPacman, to further examine their impact. By comparing these variations, we can better understand how various levels of abstraction influence learning across diverse environments.

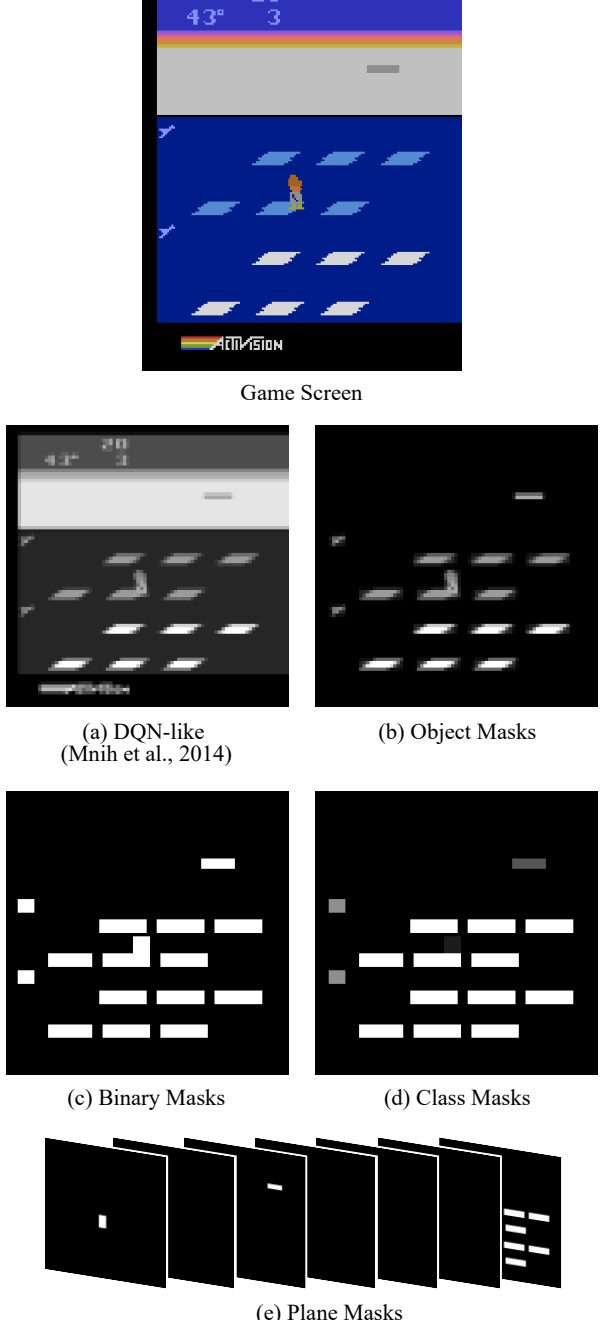

Figure 7: Alternative to Figure 2, showcasing the input representations using *Frostbite* as an example. This visualization highlights how different representations preserve or abstract various elements within the game environment.

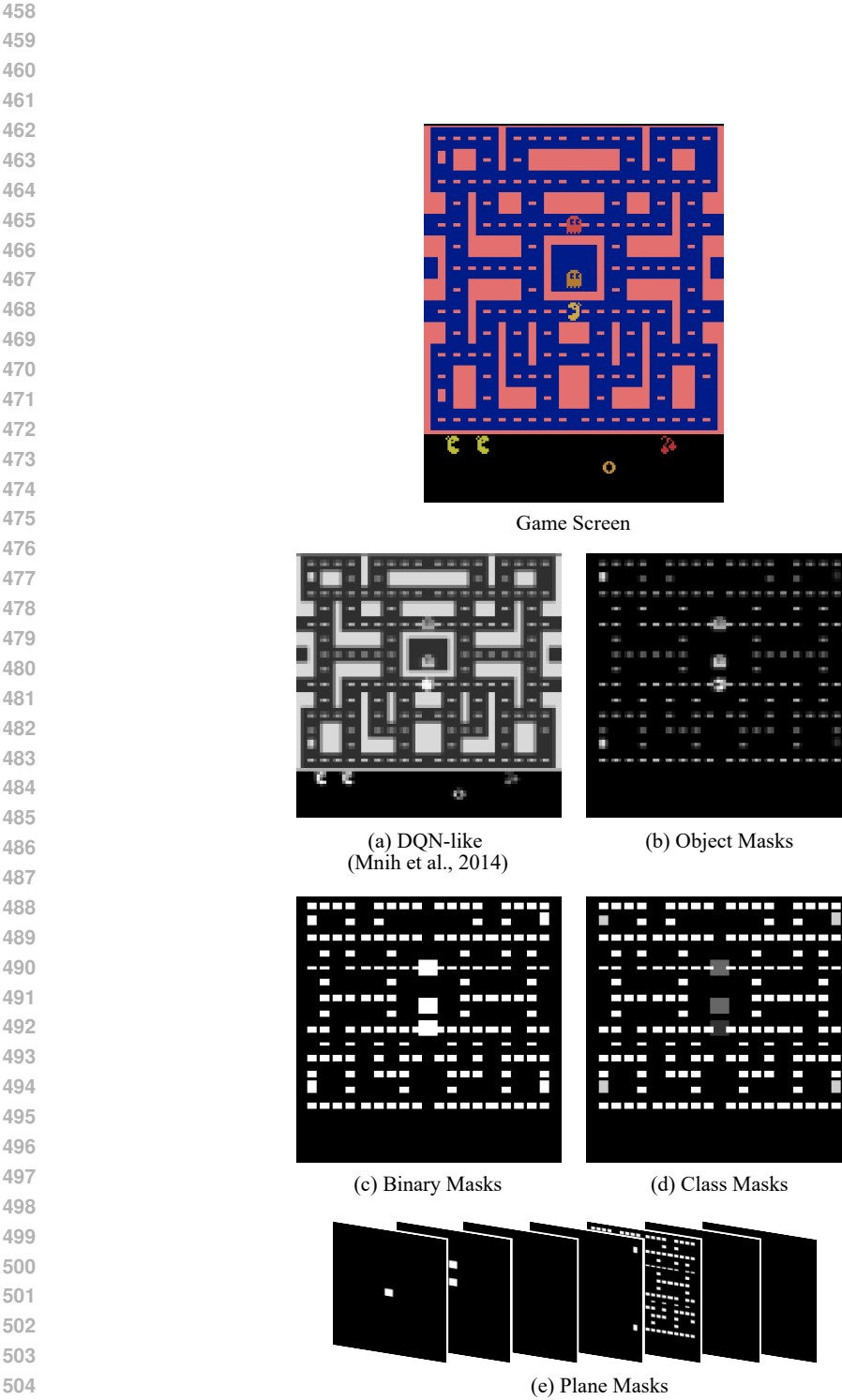

Game Screen

(a) DQN-like
(Mnih et al., 2014)

(b) Object Masks

(c) Binary Masks

(d) Class Masks

(e) Plane Masks

Figure 8: Alternative to Figure 2, showcasing the input representations using *MsPacman* as an example. This visualization highlights how different representations preserve or abstract various elements within the game environment.

# G EXTENDED EXPERIMENTAL RESULTS

To evaluate generalization, we measure how well agents trained in standard environments perform under visually and logically perturbed variants. These perturbations, drawn from the HackAtari benchmark, include visual changes (e.g., recolored objects, hidden enemies) and logical changes (e.g., modified object behavior or movement dynamics).

To provide further insight and give details about the numbers behind Figures 3 to 5, we include three tables:

- **Table 12**: Full per-environment results including our three baselines DQN, MDQN, and Semantic Vector.
- **Table 13**: Full per-environment results for PPO-based agents using all OCCAM masking variants.
- **Table 14**: The same breakdown using Rainbow as the base algorithm.

The data are used as follows:

- **Figures 3 and 4**: These figures are based on the games and variants listed in Table 12, with the data from Tables 12 to 14.
- **Figure 5**: This figure is based on the results in Tables 13 and 14.

## STATISTICAL SIGNIFICANCE

To evaluate the statistical significance of OCCAM variants relative to the DQN-like pixel baseline, we conducted Wilcoxon signed-rank tests ($n = 3$ seeds per agent) independently for each environment. We performed one-sided tests in both directions—testing whether an agent significantly outperforms or is outperformed by the baseline. Bold $p$-values in **Tables 15 and 16** indicate statistical significance ($\alpha = 0.05$) for PPO and Rainbow backbones, respectively.

The bidirectional tests reveal consistent improvements from several OCCAM variants, most notably the *Binary* and *Plane* Masks, which significantly outperform the baseline in multiple environments. In games such as *Breakout* and *Freeway*, these variants achieve $p$-values smaller than $0.1\%$ for ">  PPO/Rainbow", reflecting robust performance gains across seeds. Conversely, the Object Masks yield performance that is statistically indistinguishable from the baseline in most cases, suggesting limited benefit from object-centric abstraction alone.

To complement the per-environment tests, we additionally computed paired IQM differences across all environments, using stratified bootstrap to estimate 95% confidence intervals. These results, visualized in Figure 5, summarize the aggregate performance shift for each OCCAM variant relative to the baseline. Consistent with the Wilcoxon significance analysis, the Binary and Plane Masks exhibit the largest positive IQM shifts, while the Object Masks show minimal or no change. These paired IQM estimates provide a robust, summary-level view of performance differences across environments and further reinforce the advantage of spatially-grounded representations.

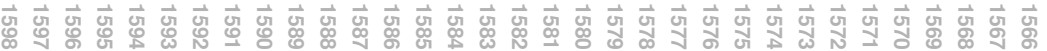

Table 12: **OCCAM improves visual robustness but struggles with logical perturbations.** This table reports the in-game episodic rewards (interquantile mean + 95% confidence interval) across a suite of 11 Atari environments, including both original and out-of-distribution variants (grey rows). Agents are trained only on the original environment and evaluated zero-shot on each variant. OCCAM-based agents (here Plane Masks) consistently outperform pixel and symbolic baselines under visual perturbations (e.g., recoloring, occlusion), demonstrating strong robustness to superficial changes. However, their performance degrades in environments with logical or structural modifications, highlighting the limitations of input-level abstraction. Best agent is highlighted in **bold**. When OCCAM masks outperform the DQN-like representation in the original game, the result is underlined. Extended and per mask results are in Appendix G, perturbation details in Appendix H.

| Game | Variant Type | PPO | | Rainbow | | Baselines | | |
|---|---|---|---|---|---|---|---|---|
| | | DQN-like Inputs | Plane Masks (Ours) | DQN-like Inputs | Plane Masks (Ours) | DQN | MDQN | Semantic Vector |
| Amidar | – | 551 [521,570] | 530 [513,542] | 381 [349,432] | 597 [451,768] | 407 [319,541] | **723** [**697,760**] | 209 [184,232] |
| Paint Roller Player | Visual Perturbation | 171 [137,207] | 524 [507,543] | 88 [70,106] | **652** [**523,782**] | 80 [62,103] | 91 [72,113] | 209 [177,234] |
| BankHeist | – | 1080 [1051,1108] | 1195 [1024,1365] | 389 [356,430] | 1126 [944,1273] | 1168 [1123,1208] | **1406** [**1312,1488**] | – |
| Random City | Logical Perturbation | 1105 [1076,1122] | 1179 [1032,1325] | 384 [362,419] | 1066 [915,1209] | 1047 [986,1118] | 1080 [1050,1108] | – |
| Bowling | – | 65 [64,67] | 62 [60,65] | **82** [**80,84**] | 69 [63,78] | 28 [22,34] | 32 [30,34] | 62 [60,65] |
| Shift Player | Logical Perturbation | **63** [**61,66**] | 51 [31,64] | 48 [20,76] | 58 [40,75] | 30 [22,35] | 23 [18,28] | 62 [60,65] |
| Boxing | – | 95 [94,96] | 96 [94,98] | **97** [**96,98**] | 91 [87,96] | 90 [86,93] | **97** [**96,98**] | 93 [90,96] |
| Boxers Red and Blue | Visual Perturbation | 4 [1,8] | **98** [**97,99**] | 1 [−1,5] | 92 [87,96] | −2 [−4,−0] | −1 [−2,1] | 91 [89,93] |
| Breakout | – | 131 [84,183] | **280** [**209,338**] | 54 [37,85] | 86 [47,142] | 56 [35,76] | 168 [121,224] | 37 [30,43] |
| Player and Ball Red | Visual Perturbation | 7 [6,10] | **315** [**261,358**] | 7 [4,11] | 57 [31,109] | 5 [4,6] | 27 [22,35] | 35 [27,42] |
| Freeway | – | 32 [31,33] | 33 [33,34] | **34** [**34,34**] | 33 [33,34] | 27 [16,33] | **34** [**34,34**] | 31 [31,31] |
| Stop All Cars | Logical Perturbation | 8 [0,20] | 22 [6,37] | 0 [0,0] | 8 [0,20] | 0 [0,0] | **33** [**33,34**] | 0 [0,0] |
| All Cars Black | Visual Perturbation | 24 [22,25] | 33 [33,34] | 25 [24,26] | **33** [**32,33**] | 12 [9,14] | 25 [25,26] | 31 [31,31] |
| Frostbite | – | 300 [292,306] | 282 [275,290] | 2749 [2374,3140] | 3254 [2869,3632] | 3445 [3012,3766] | **4349** [**2554,5791**] | 265 [258,390] |
| Static Ice | Logical Perturbation | 40 [40,40] | 25 [0,72] | **684** [**214,1445**] | 40 [14,81] | 42 [22,71] | 26 [7,64] | 30 [0,82] |
| MsPacman | – | 3001 [2803,3396] | **6130** [**5603,6681**] | 3004 [2701,3552] | 3854 [3592,4167] | 2469 [2326,2614] | 2406 [2269,2560] | 1919 [1406,2631] |
| 2nd Level | Logical Perturbation | 97 [60,184] | 334 [269,407] | **1079** [**911,1271**] | 698 [626,775] | 469 [376,581] | 384 [329,461] | 72 [60,80] |
| Pong | – | 15 [14,16] | 19 [18,19] | 18 [17,19] | **20** [**20,20**] | 18 [17,19] | 17 [16,18] | 19 [18,20] |
| Lazy Enemy | Logical Perturbation | −8 [−11,−6] | **18** [**17,18**] | −4 [−8,−0] | 16 [15,17] | −2 [−4,−0] | −5 [−8,−1] | −19 [−21,−16] |
| Riverraid | – | 7668 [7279,8046] | 8020 [7824,8216] | 3009 [2203,4161] | 3043 [2714,3396] | **9202** [**8895,9649**] | 8633 [8201,9055] | 3306 [3069,3567] |
| Color Change | Visual Perturbation | 441 [262,772] | **8263** [**8026,8504**] | 509 [432,600] | 3546 [3118,4086] | 285 [210,391] | 868 [754,1034] | 3525 [3256,3786] |
| Linear River | Logical Perturbation | 6932 [5770,7722] | 1969 [1574,2406] | 5018 [3754,6349] | 3282 [3191,3375] | 10901 [9654,12311] | **11568** [**10666,12444**] | 356 [301,388] |
| SpaceInvaders | – | 744 [703,796] | 1556 [1267,1892] | 1003 [886,1185] | 972 [823,1174] | 1248 [1096,1389] | **1759** [**1473,2078**] | 358 [308,419] |
| Shields off by 3 | Logical Perturbation | 621 [558,683] | **1429** [**1061,1789**] | 710 [571,899] | 987 [840,1164] | 564 [478,670] | 729 [538,927] | 415 [342,485] |

Table 13: This table compares the average episodic rewards (interquartile mean + 95% confidence interval) achieved by *PPO* using different input representations: the standard DQN-like representation, OCAtari's Semantic Vector, and our structured masking approaches, including Object Masks, Binary Masks, Class Masks, and Plane Masks. Results are averaged over three random seeds, with standard deviations indicating variability. All agents were trained in the original game environment and evaluated in the standard setting and visually or behaviorally perturbed variations (gray-shaded). The table highlights how different levels of abstraction in input representations impact the agents' performance and robustness across diverse environments. These results are normalized (using HNS) and visualized in Figure 5.

| Game | Variant Type | PPO | Object Masks | Binary Masks | Class Masks | Plane Masks | Semantic Vector |
|---|---|---|---|---|---|---|---|
| Amidar | – | 551 [521,570] | 566 [516,619] | 530 [426,610] | 499 [471,523] | 530 [513,542] | 209 [184,232] |
| Paint Roller Player | Visual Perturbation | 171 [137,207] | 332 [264,386] | 526 [436,592] | 496 [447,540] | 524 [507,543] | 209 [177,234] |
| Enemy to Pig | Visual Perturbation | 555 [542,565] | 346 [301,390] | 544 [443,616] | 507 [478,530] | 101 [84,118] | 358 [88,748] |
| BankHeist | – | 1080 [1051,1108] | 794 [782,806] | 1221 [1206,1234] | 1193 [1162,1215] | 1195 [1024,1365] | – |
| Random City | Logical Perturbation | 1105 [1077,1122] | 793 [782,808] | 1226 [1206,1236] | 1216 [1188,1229] | 1179 [1032,1324] | – |
| Bowling | – | 65 [64,67] | 66 [63,68] | 62 [60,64] | 68 [65,70] | 62 [60,65] | 62 [60,65] |
| Shift Player | Logical Perturbation | 63 [61,66] | 53 [37,64] | 62 [60,64] | 66 [63,68] | 51 [31,64] | 62 [60,65] |
| Boxing | – | 95 [94,96] | 94 [92,95] | 96 [94,97] | 92 [90,93] | 96 [94,98] | 93 [90,96] |
| Boxers Red and Blue | Visual Perturbation | 4 [1,8] | 79 [76,81] | 96 [95,97] | 92 [90,94] | 98 [97,99] | 91 [89,93] |
| Breakout | – | 131 [84,183] | 43 [26,82] | 166 [105,227] | 101 [55,160] | 280 [209,338] | 37 [30,43] |
| Player and Ball Red | Visual Perturbation | 7 [6,10] | 17 [10,35] | 127 [83,186] | 189 [139,232] | 315 [261,358] | 35 [27,42] |
| All Blocks Red | Visual Perturbation | 156 [109,200] | 126 [70,190] | 194 [142,244] | 163 [117,208] | 194 [129,255] | 30 [23,37] |
| Freeway | – | 32 [31,33] | 33 [33,34] | 33 [33,34] | 33 [33,33] | 33 [33,34] | 31 [31,31] |
| Stop All Cars | Logical Perturbation | 8 [0,20] | 0 [0,0] | 0 [0,0] | 7 [0,19] | 22 [6,37] | 0 [0,0] |
| All Cars Black | Visual Perturbation | 24 [22,25] | 22 [21,24] | 33 [33,34] | 33 [33,33] | 33 [33,34] | 31 [31,31] |
| Frostbite | – | 300 [292,306] | 282 [270,295] | 298 [285,310] | 259 [250,266] | 282 [275,290] | 265 [258,390] |
| Static Ice | Logical Perturbation | 40 [40,40] | 14 [0,44] | 1 [0,9] | 1 [0,10] | 25 [0,72] | 30 [0,82] |
| MsPacman | – | 3001 [2803,3396] | 5311 [4877,5690] | 4632 [4194,5038] | 4038 [3591,4449] | 6130 [5603,6681] | 1919 [1406,2631] |
| 2nd Level | Logical Perturbation | 97 [60,184] | 302 [261,352] | 639 [466,831] | 419 [273,584] | 334 [269,407] | 72 [60,80] |
| Pong | – | 15 [14,16] | 18 [18,19] | 19 [18,20] | 19 [19,20] | 19 [18,19] | 19 [18,20] |
| Lazy Enemy | Logical Perturbation | −8 [−11,−6] | 14 [12,15] | 11 [4,16] | −1 [−6,4] | 18 [17,18] | −19 [−21,−16] |
| Hidden Enemy | Visual Perturbation | – | 17 [16,18] | 19 [18,20] | 19 [19,20] | 19 [18,20] | −19 [−21,−16] |
| Riverraid | – | 7668 [7279,8046] | 7784 [7659,7916] | 7908 [7739,8073] | 7714 [7524,7915] | 8020 [7824,8216] | 3306 [3069,3567] |
| Color Change | Visual Perturbation | 441 [262,772] | 7831 [7668,8014] | 7812 [7676,7980] | 7631 [7464,7801] | 8263 [8026,8504] | 3525 [3256,3786] |
| Linear River | Logical Perturbation | 6932 [5770,7722] | 2878 [2750,3130] | 2698 [2690,2712] | 2885 [2724,3146] | 1969 [1574,2406] | 356 [301,388] |
| SpaceInvaders | – | 744 [703,796] | 645 [579,705] | 794 [710,875] | 463 [402,525] | 1556 [1267,1892] | 358 [308,419] |
| Shields off by 1 | Logical Perturbation | 761 [707,797] | 632 [574,692] | 597 [515,678] | 337 [260,426] | 1608 [1235,2064] | 318 [280,366] |
| Shields off by 3 | Logical Perturbation | 621 [558,683] | 583 [539,638] | 473 [375,566] | 335 [275,428] | 1429 [1061,1789] | 415 [342,485] |

Table 14: This table compares the average episodic rewards (interquartile mean + 95% confidence interval) achieved by *Rainbow* using different input representations: the standard DQN-like representation, and our structured masking approaches, including Object Masks, Binary Masks, Class Masks, and Plane Masks. As an additional baseline, we added MDQN (Vieillard et al., 2020). Results are averaged over three random seeds, with confidence intervals indicating variability. All agents were trained in the original game environment and evaluated in the standard setting, as well as visually or behaviorally perturbed variations (gray-shaded). The table highlights how different levels of abstraction in input representations impact the agents' performance and robustness across diverse environments. These results are normalized (using HNS) and visualized in Figure 5.

| Game | Variant Type | Rainbow | Object Masks | Binary Masks | Class Masks | Plane Masks | MDQN |
|---|---|---|---|---|---|---|---|
| Amidar | – | 381 [349,432] | 894 [796,981] | 632 [596,662] | 541 [443,643] | 597 [451,768] | 723 [697,760] |
| Paint Roller Player | Visual Perturbation | 88 [70,106] | 34 [24,47] | 640 [574,682] | 634 [532,720] | 652 [523,782] | 91 [72,113] |
| Enemy to Pig | Visual Perturbation | 645 [533,786] | 130 [94,184] | 654 [615,695] | 489 [346,647] | 323 [248,432] | 740 [596,886] |
| BankHeist | – | 389 [356,430] | 597 [518,658] | 873 [780,1007] | 1012 [879,1147] | 1126 [944,1273] | 1406 [1312,1488] |
| Random City | Logical Perturbation | 384 [362,419] | 603 [504,676] | 849 [758,993] | 944 [828,1096] | 1066 [916,1208] | 1411 [1319,1481] |
| Bowling | – | 82 [80,84] | 68 [64,70] | 101 [80,132] | 55 [45,62] | 69 [63,78] | 32 [30,34] |
| Shift Player | Logical Perturbation | 48 [20,76] | 26 [17,40] | 68 [67,69] | 32 [20,47] | 58 [40,75] | 23 [18,28] |
| Boxing | – | 97 [96,98] | 94 [93,96] | 91 [84,96] | 92 [92,92] | 55 [39,67] | 97 [96,98] |
| Boxers Red and Blue | Visual Perturbation | 1 [−1,5] | −8 [−11,−5] | 92 [87,96] | 92 [92,92] | 56 [41,68] | −1 [−2,1] |
| Breakout | – | 54 [37,85] | 70 [56,108] | 197 [125,272] | 51 [36,68] | 86 [47,142] | 168 [121,224] |
| Player and Ball Red | Visual Perturbation | 7 [4,11] | 7 [5,9] | 171 [104,255] | 48 [32,69] | 57 [31,109] | 27 [22,35] |
| All Blocks Red | Visual Perturbation | 83 [52,140] | 105 [66,171] | 113 [80,161] | 50 [35,70] | 38 [21,99] | 304 [252,340] |
| Freeway | – | 34 [34,34] | 33 [32,34] | 34 [34,34] | 34 [34,34] | 33 [33,34] | 34 [34,34] |
| Stop All Cars | Logical Perturbation | 0 [0,0] | 7 [0,20] | 33 [20,41] | 0 [0,0] | 8 [0,20] | 34 [34,34] |
| All Cars Black | Visual Perturbation | 25 [24,26] | 33 [32,34] | 34 [34,34] | 34 [34,34] | 33 [32,33] | 25 [25,26] |
| Frostbite | – | 2749 [2374,3140] | 3426 [3240,3653] | 3801 [3453,4219] | 3604 [3317,3868] | 3254 [2869,3632] | 4349 [2554,5791] |
| Static Ice | Logical Perturbation | 684 [214,1445] | 2544 [1894,2996] | 417 [126,1022] | 885 [459,1442] | 40 [14,81] | 26 [7,64] |
| MsPacman | – | 3004 [2701,3552] | 3731 [3297,4171] | 2744 [2352,3127] | 2962 [2482,3604] | 3854 [3592,4167] | 2406 [2269,2560] |
| 2nd Level | Logical Perturbation | 1079 [911,1271] | 791 [646,934] | 490 [432,569] | 571 [486,664] | 698 [626,775] | 384 [329,461] |
| Pong | – | 18 [17,19] | 21 [20,21] | 20 [19,20] | 21 [20,21] | 20 [20,20] | 17 [16,18] |
| Lazy Enemy | Logical Perturbation | −4 [−8,−0] | −9 [−15,−1] | −7 [−14,2] | −19 [−20,−17] | 16 [15,17] | −5 [−8,−1] |
| Hidden Enemy | Visual Perturbation | – | 19 [18,20] | 18 [17,19] | 18 [16,20] | 19 [19,19] | – |
| Riverraid | – | 3009 [2203,4161] | 2482 [2482,2482] | 3383 [2834,4069] | 2888 [2524,3131] | 3043 [2714,3396] | 8633 [8201,9055] |
| Color Change | Visual Perturbation | 509 [432,600] | 2288 [2288,2288] | 3544 [3344,3874] | 3144 [2639,3526] | 3546 [3118,4086] | 868 [754,1034] |
| Linear River | Logical Perturbation | 5018 [3754,6349] | 2508 [2508,2508] | 3269 [3132,3379] | 2998 [2559,3154] | 3282 [3191,3375] | 11568 [10666,12444] |
| SpaceInvaders | – | 1003 [886,1185] | 420 [351,493] | 713 [521,925] | 724 [647,820] | 972 [823,1174] | 1759 [1473,2078] |
| Shields off by 1 | Logical Perturbation | 981 [791,1191] | 368 [288,459] | 533 [450,644] | 628 [538,737] | 951 [756,1152] | 1827 [1506,2148] |
| Shields off by 3 | Logical Perturbation | 710 [571,899] | 352 [295,423] | 606 [523,673] | 403 [324,495] | 987 [840,1164] | 729 [538,927] |

Table 15: Wilcoxon signed-rank test results in both directions for each OCCAM + PPO variant compared to the DQN-like representation. Each row reports the results of two one-sided Wilcoxon signed-rank tests comparing an OCCAM variant to the PPO baseline in a given environment ($n = 3$ seeds). The left value in each pair (labeled "> PPO") tests whether the OCCAM variant significantly outperforms PPO; the right value (labeled "< PPO") tests whether PPO significantly outperforms the OCCAM variant. **Bold** values indicate a statistical significance with $\alpha = 0.05$.

| Environment | Binary Masks | | Class Masks | | Object Masks | | Plane Masks | |
|---|---|---|---|---|---|---|---|---|
| | > PPO | < PPO | > PPO | < PPO | > PPO | < PPO | > PPO | < PPO |
| Amidar | 0.7002 | 0.2998 | 0.9850 | **0.0150** | 0.0923 | 0.9077 | 0.8201 | 0.1799 |
| BankHeist | **0.0000** | 1.0000 | **0.0000** | 1.0000 | 1.0000 | **0.0000** | **0.0189** | 0.9811 |
| Bowling | 0.9999 | **0.0001** | **0.0273** | 0.9727 | 0.6805 | 0.3195 | 0.9469 | 0.0531 |
| Boxing | 0.1479 | 0.8521 | 0.9984 | **0.0016** | 0.8372 | 0.1628 | 0.1316 | 0.8684 |
| Breakout | 0.2053 | 0.7947 | 0.6649 | 0.3425 | 0.9598 | **0.0420** | **0.0030** | 0.9970 |
| Freeway | **0.0001** | 0.9999 | **0.0022** | 0.9978 | **0.0000** | 1.0000 | **0.0000** | 1.0000 |
| Frostbite | 0.6265 | 0.3735 | 1.0000 | **0.0000** | 0.9998 | **0.0002** | 1.0000 | **0.0000** |
| MsPacman | **0.0002** | 0.9998 | **0.0012** | 0.9989 | **0.0000** | 1.0000 | **0.0000** | 1.0000 |
| Pong | **0.0000** | 1.0000 | **0.0000** | 1.0000 | **0.0002** | 0.9998 | **0.0001** | 0.9999 |
| Riverraid | 0.0841 | 0.9159 | 0.2789 | 0.7211 | 0.1424 | 0.8576 | **0.0303** | 0.9712 |
| SpaceInvaders | 0.2390 | 0.7610 | 1.0000 | **0.0000** | 0.9610 | **0.0390** | 0.0000 | 1.0000 |
| Count | **4** | 1 | **5** | 4 | 3 | **4** | **7** | 1 |

Table 16: Wilcoxon signed-rank test results comparing each OCCAM + Rainbow variant to the Rainbow (DQN-like) baseline across Atari environments ($n = 3$ seeds per agent). Each cell shows the one-sided $p$-value for the indicated test direction. **Bold** values indicate statistical significance ($\alpha = 0.05$).

| Environment | Binary Masks | | Class Masks | | Object Masks | | Plane Masks | |
|---|---|---|---|---|---|---|---|---|
| | > Rainbow | < Rainbow | > Rainbow | < Rainbow | > Rainbow | < Rainbow | > Rainbow | < Rainbow |
| Amidar | **0.0010** | 1.0000 | 0.4229 | 0.6152 | **0.0010** | 1.0000 | **0.0010** | 1.0000 |
| BankHeist | **0.0010** | 1.0000 | **0.0010** | 1.0000 | **0.0010** | 1.0000 | **0.0010** | 1.0000 |
| Bowling | **0.0010** | 1.0000 | 1.0000 | **0.0010** | 1.0000 | **0.0010** | 1.0000 | **0.0010** |
| Boxing | **0.0020** | 1.0000 | 0.9746 | **0.0293** | 0.8125 | 0.2051 | 1.0000 | **0.0010** |
| Breakout | 0.1016 | 0.9180 | 0.7842 | 0.2461 | 0.4717 | 0.5498 | 0.2158 | 0.8125 |
| Freeway | 0.9375 | 0.3125 | 0.2500 | 1.0000 | 0.8125 | 0.5000 | 0.7500 | 0.5000 |
| Frostbite | 0.1377 | 0.8838 | **0.0322** | 0.9756 | **0.0322** | 0.9756 | **0.0244** | 0.9814 |
| MsPacman | 0.1006 | 0.9072 | 0.7842 | 0.2461 | **0.0244** | 0.9814 | **0.0420** | 0.9678 |
| Pong | 0.1543 | 0.8652 | 0.0684 | 0.9395 | **0.0039** | 1.0000 | 0.0547 | 0.9609 |
| Riverraid | **0.0029** | 0.9980 | 0.3262 | 0.7148 | 0.1826 | 0.8330 | 0.0967 | 0.9199 |
| SpaceInvaders | 0.9727 | **0.0371** | 0.9668 | **0.0410** | 0.9990 | **0.0020** | 0.4229 | 0.6152 |
| Count | **5** | 1 | 2 | **3** | **5** | 1 | **4** | 2 |

Table 17: **Combining Masks and DQN-like representation result in improved performance while keeping the robustness gain of the masks.** We investigated the combination of our input representation with the DQN-like baseline in 3 games. Setup is the same as and results are comparable to Table 14 (PPO, 3 seeds, 40M frames).

| Game (Variation) | Object Masks+Pixels | Binary Masks+Pixels | Class Masks+Pixels | Plane Masks+Pixels |
|---|---|---|---|---|
| Riverraid | 7847 [7590,8173] | 8381 [8210,8549] | 7823 [7517,8273] | 9753 [9523,10007] |
| *Linear River* | 7820 [7631,8065] | 8389 [8144,8642] | 7967 [7655,8423] | 9603 [9264,9874] |
| *Color Change* | 7948 [7750,8072] | 8502 [8319,8634] | 7827 [7555,8300] | 9623 [9338,9833] |
| MsPacman | 6519 [5725,7374] | 4656 [4182,5203] | 5530 [4875,6064] | 5931 [5424,6571] |
| *Level 2* | 457 [337,606] | 452 [369,579] | 636 [441,858] | 170 [92,299] |
| Freeway | 33 [33,33] | 33 [33,33] | 33 [33,33] | 33 [33,33] |
| *All Cars Black* | 26 [26,26] | 33 [33,33] | 33 [33,33] | 34 [33,34] |
| *Stop All Cars* | 33 [33,34] | 33 [33,33] | 33 [33,33] | 34 [33,33] |

MASKS+PIXELS: COMBINING MASKS WITH DQN-LIKE REPRESENTATION

We have seen that over-abstraction can result in failures, as observed in environments like Riverraid (cf. Table 3). To investigate whether we can mitigate the issue of over-abstraction, we conduct a small ablation study that augments OCCAM masks with the DQN-like pixel input. The goal is to test whether reintroducing global visual context, while still providing structured object information, can recover performance without sacrificing robustness. Specifically, we concatenate the mask (Binary, Class, or Plane Masks) with the standard DQN-like grayscale stack, enabling the agent to access both object-centric structure and full-scene layout. This experiment investigates whether Masks+Pixels mitigates performance drops caused by removing excessive information, and how this hybrid representation affects nominal performance and robustness under perturbations. The results are shown in Table 17: Masks+Pixels improves nominal performance in some games (Planes in Riverraid and 3 out of 4 masks in MsPacman), while in the rest, performance is similar. Testing them in the perturbations also showed that the robustness gained with masks against, e.g., color changes still persists even when the colors are presented in the inputs (Freeway, Riverraid). This is interesting, letting us keep the best of both while only doubling the input size (Binary, Object, Class Masks) or adding one additional plane (Plane Masks). A deeper analysis of how policies actually use the different channels (e.g., via attribution or representation probing) is an interesting direction.

TRAINING MASKS: PRELIMINARY MADI RESULTS

We conducted a preliminary evaluation of MaDi (Grooten et al., 2024) as a learned masking baseline. Since MaDi was originally developed with continuous-action environments in mind and paired with SAC, we adapted it to the discrete-action Atari setting by replacing SAC with the same PPO backbone used in our baselines and inserting the MaDi Masker module in front of the policy. All hyperparameters were kept consistent with our PPO experiments (40M frames, 3 seeds; Appendix B), and no additional image augmentation was applied.

Under this setup, MaDi showed little to no sustainable learning in Atari, underperforming PPO and Rainbow on clean environments across 10 of 11 games (see Figure 9). We attribute this to several factors: (i) *sensitivity to reward scale and optimization*: the Masker's gradient signal can vanish or explode under Atari's sparse and delayed rewards; (ii) *tuning requirements*: the Masker introduces additional hyperparameters that likely require environment-specific adjustment; and (iii) *reward density*: MaDi relies on dense reward signals to provide meaningful guidance for masking, a property not present in Atari benchmarks. Together, these challenges make MaDi less suited to discrete, sparse-reward environments without substantial adaptation.

For transparency, Table 18 lists the hyperparameters used and Figure 9 shows per-game learning curves. A more thorough tuning study, including image augmentation and alternative optimizers, is left to future work.

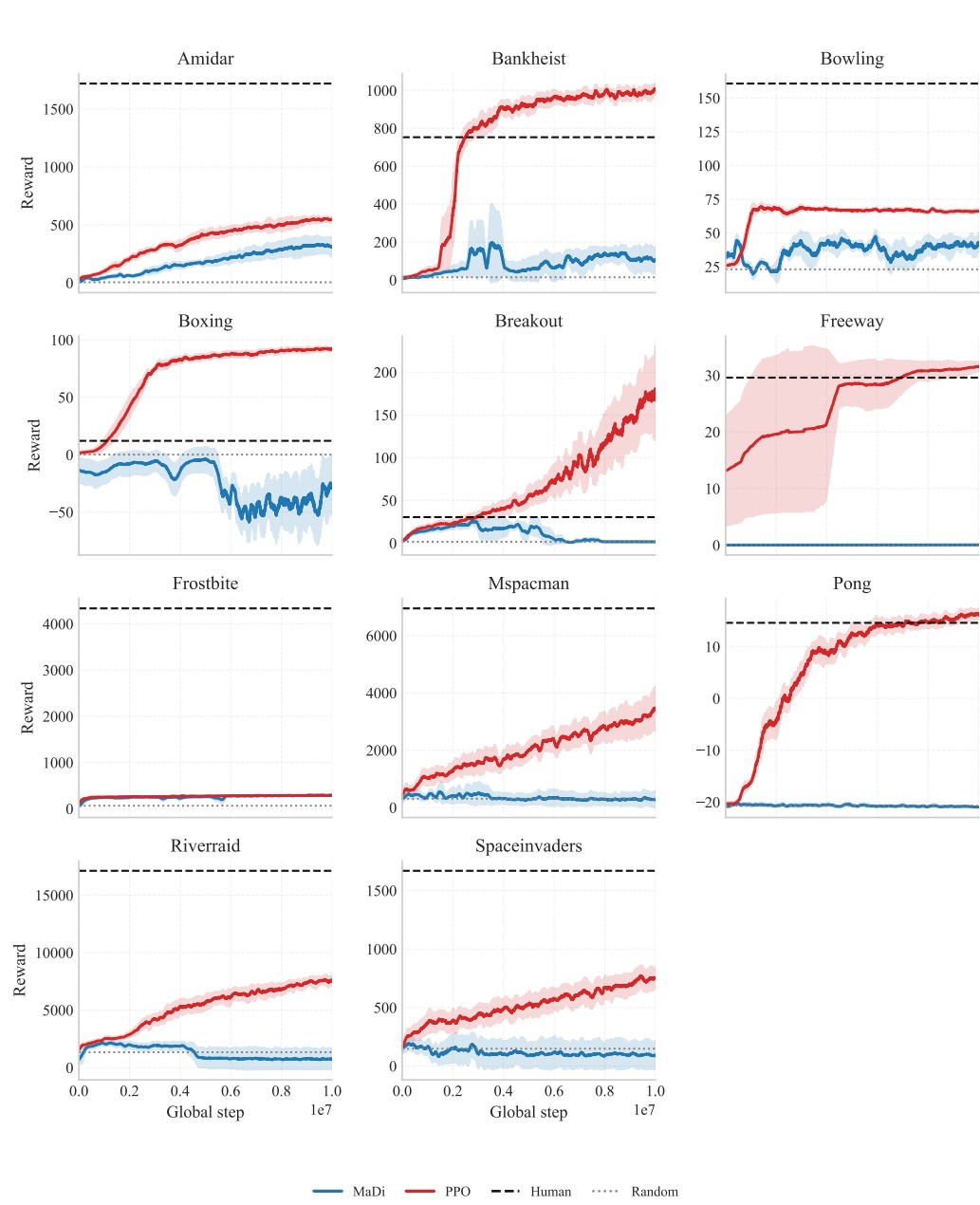

Figure 9: **MaDi fails to learn across most Atari games.** Average episodic return over 3 seeds for MaDi trained with the same hyperparameters as PPO (see Appendix B). MaDi fails to make progress in 10 out of 11 games, highlighting that learned masking under this setup is substantially less effective than our baseline PPO agent.

Table 18: Key hyperparameters used for our **MaDi** training, following Grooten et al. (2024).

| Hyperparameter | Value |
|---|---|
| Masker Learning Rate | $10^{-3}$ |
| Masker $\beta$ | 0.9 |
| Number of Layers | 3 |
| Number of Filters | 32 |
| Image Augmentation | `False` |

## H   ATARI GAME VARIANTS

Below, we briefly describe each Atari game variant used in our study. Descriptions are done by us or taken from the ALE Documentation[4]. These variants are visualized in Figure 10 and created using the HackAtari Environment (Delfosse et al., 2025). The modification list to create each variant is given in Table 19. This can be used to visualize or evaluate the models' performances.

### HACKATARI

To create any of the following variants for a specific Atari game (`env_id`), you can simply pass the modification list (`modifs`) for the variant into HackAtari:

```
env = HackAtari(env_id, modifs=modifs)
```

### AMIDAR VARIANTS

**Amidar:** *Amidar* is similar to Pac-Man: You are trying to visit all places on a 2-dimensional grid while simultaneously avoiding your enemies.

**Enemy to Pig:** Change the enemy warriors into Pigs. The game logic stays the same.

**Player to Roller:** Changing the sprite of the player figure from a human to a paint roller. The game logic stays the same.

### BANKHEIST VARIANTS

**BankHeist:** You are a bank robber and (naturally) want to rob as many banks as possible. You control your getaway car and must navigate maze-like cities. The police chase you and will appear whenever you rob a bank. You may destroy police cars by dropping sticks of dynamite. You can fill up your gas tank by entering a new city. This is your *BankHeist*.

**Random City:** The starting city and the city you enter are randomly selected and do not follow a pattern.

### BOWLING VARIANTS

**Bowling:** Your goal is to score as many points as possible in the game of *Bowling*. A game consists of 10 frames, and you have two tries per frame.

**Shift Player:** The player starts closer to the pins.

### BOXING VARIANTS

**Boxing:** The standard *Boxing* environment where two players compete to land punches.

**Boxers Red and Blue:** A modified version where one player is red and the other is blue, potentially influencing object perception.

---

[4]`https://ale.farama.org/`

### BREAKOUT VARIANTS

**Breakout:** The original *Breakout* environment where the player controls a paddle to break bricks.

**All Blocks Red:** All blocks are red, removing the color of blocks which does not hold game-relevant information.

**Player and Ball Red:** The paddle is slightly redder than usual, potentially changing agent perception and behavior.

### FREEWAY VARIANTS

**Freeway:** The standard *Freeway* environment where a chicken crosses a highway with moving cars.

**All Cars Black:** All vehicles are black, reducing visual diversity between the cars. There are no additional changes.

**Stop All Cars:** All cars are stopped on the edge of the frame, making it trivial to pass the street.

### FROSTBITE VARIANT

**Frostbite:** The original *Frostbite* environment where the player builds an igloo while avoiding hazards. To collect ice, the player has to jump between moving ice shelves.

**Static Ice:** Ice platforms remain fixed instead of moving, altering difficulty (making it much easier).

### MSPACMAN VARIANTS

**MsPacman** The standard *MsPacman* environment that is very similar to the Pac-Man environment.

**2nd Level:** A later level with a changed maze structure. No agent ever reached this level in training. As such, it is out of distribution for all agents.

### PONG VARIANTS

**Pong:** The standard *Pong* environment where two paddles hit a ball back and forth.

**Lazy Enemy:** The opponent stays still while the ball is flying away from it and only starts moving after the player hits the ball. No visual changes are visible. This was presented as one of two examples of misalignment in Delfosse et al. (2024b).

**Hidden Enemy:** The opponent is hidden for the player. The only observable objects are the paddle and the Ball. This follows the experiment by Delfosse et al. (2024b) and is used to compare to their work directly.

### RIVERRAID VARIANTS

**Riverraid:** In *Riverraid*, you control a jet that flies over a river: you can move it sideways and fire missiles to destroy enemy objects. Each time an enemy object is destroyed, you score points (i.e., rewards).

**Color Change:** Change the color of all objects to a different preset color—no change in the game logic.

**Linear River:** Fix the river to always have the same shape—a single line—and width: no splits, etc.

### SPACEINVADERS VARIANTS

**SpaceInvaders:** In *SpaceInvaders*, your objective is to destroy the space invaders by shooting your laser cannon at them before they reach the Earth. The game ends when all your lives are lost after taking enemy fire or when they reach the earth.

**Shields off by X:** The shields are moved X pixel(s) to the right. No further changes.

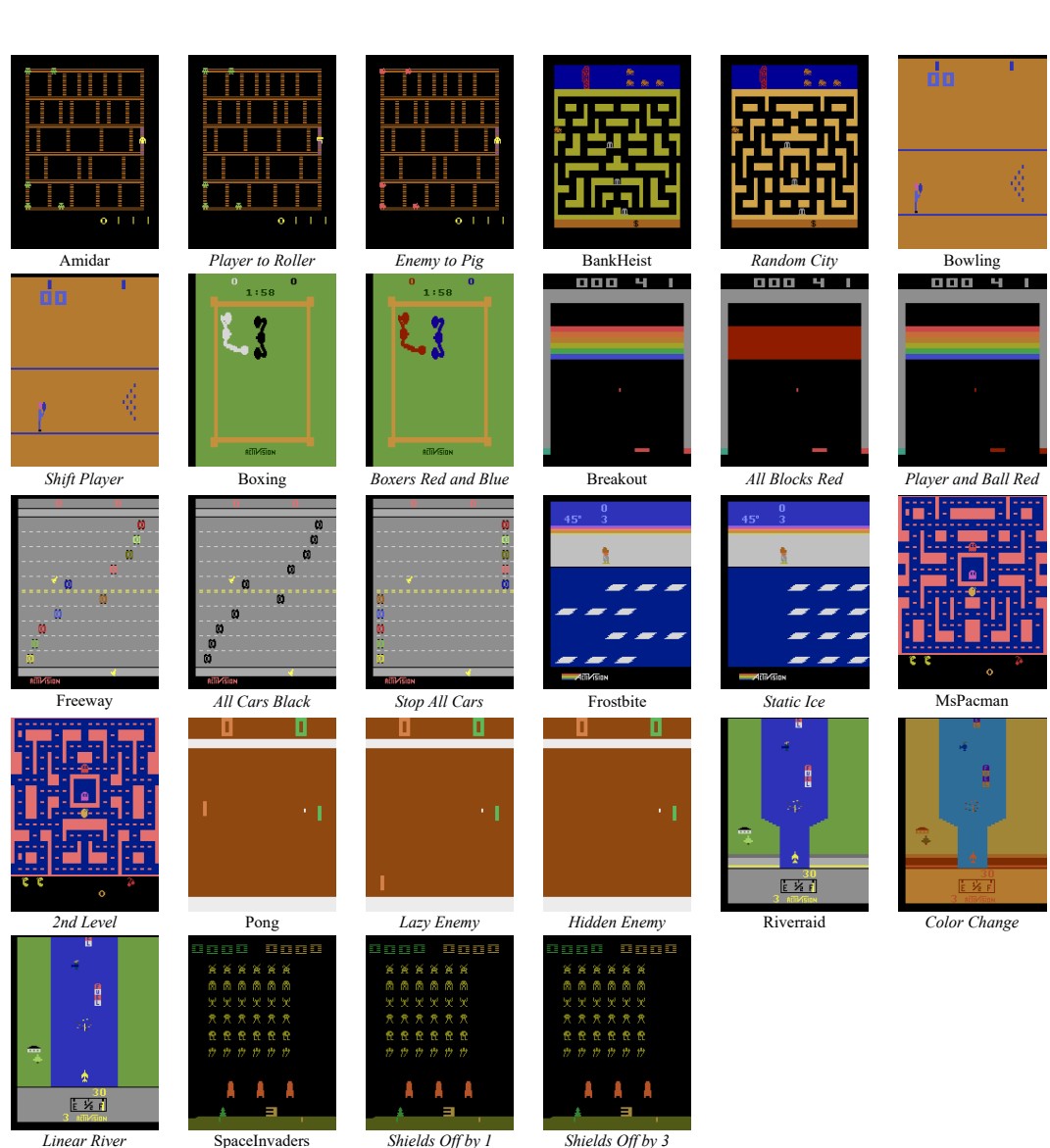

Figure 10: Illustration of different game variants used in our study. Games are grouped, starting with the original game on the left, followed by the variants (*italic* label). Groups may break to the next row. These variants introduce modifications such as color changes and (enemy) behavior variations to study their impact on learning and generalization.

Table 19: The HackAtari modifications for all game variants used in our work.

| **Amidar** | |
|---|---|
| Enemy to Pig | `pig_enemies` |
| Player to Roller | `paint_roller_player` |
| **BankHeist** | |
| Random City | `random_city` |
| **Boxing** | |
| Boxers Red and Blue | `color_player_red`, `color_enemy_blue` |
| **Breakout** | |
| All Blocks Red | `color_all_blocks_red` |
| Player and Ball Red | `color_player_and_ball_red` |
| **Bowling** | |
| Shift Player | `shift_player` |
| **Freeway** | |
| All Cars Black | `all_black_cars` |
| Stop All Cars | `stop_all_cars_edge` |
| **Frostbite** | |
| Static Ice | `reposition_floes_easy` |
| **MsPacman** | |
| Level 2 | `set_level_1` |
| **Pong** | |
| Lazy Enemy | `lazy_enemy` |
| Hidden Enemy | `hidden_enemy` |
| **Riverraid** | |
| Color Change | `game_color_change01` |
| Linear River | `linear_river` |
| **SpaceInvaders** | |
| Shields off by 1 | `relocate_shields_off_by_one` |
| Shields off by 3 | `relocate_shields_off_by_three` |

