# OpenReview forum: "Plug-and-Play Object-Centric Reinforcement Learning via Masking"
_ICLR.cc/2026/Conference — ICLR 2026 Conference Desk Rejected Submission_

### Official Review · Reviewer_8ZNP · 2025-10-15

**Soundness:** 3
**Presentation:** 3
**Contribution:** 1
**Rating:** 4
**Confidence:** 5

**Summary:**

The paper studies the influence of utilizing object masks as object-centric inductive bias for deep RL from pixels, in the context of environment generalization (visual and dynamics). The proposed method, OCCAM, is a simple enhancement where supervised/ground-truth object masks, of various types (where “Plane Masks” is the best-performing one), are provided to the agent and integrated in the standard CNN-based learning pipeline. The method is evaluated in terms of generalization to some visual perturbations (where it demonstrates robustness) and environment dynamics perturbations.

**Strengths:**

* Simple.
* The paper reads well and is easy to follow.
* Well-detailed appendix!
* Thorough analysis on Atari.
* The method demonstrates some robustness to visual perturbations.
* Open-source code.

**Weaknesses:**

* Relies on supervision, i.e., assumes access to segmentations and detections which are trained with supervision or acquired from the environment (only possible in simulation). This kind of approach must assume that the pre-trained detector can, in fact, detect the objects in the environment, which is not typically true, especially given domain distribution-shifts. Alternatively, one can train a supervised detector in the environment, but this requires additional human effort. This reliance weakens the method and contribution in my opinion.
* Background: there is an underlying assumption that background features do not matter (and please correct me if I misunderstood). However, I don’t agree that this is generally true (as demonstrated in the *Riverraid* game), as agents should be spatially-aware (e.g, obstacles or walls in a maze environment). There is also a claim that masking removes irrelevant task features like textures and colors, but what if appearance is important to the task? If this claim only concerns visual perturbations of the background, and the background is not relevant to the task, then I somewhat agree, but otherwise, I don’t think this is generally true.
* The main goal of the paper is to evaluate the contribution of masking. Given previous work has already shown that object-centric representations are indeed outperforming traditional approaches, the results and findings in the paper are not entirely convincing, leading me to think that the policy design in this work is lacking and not fitted for object-centric representations (see [Zhou, Allan, et al. "Policy architectures for compositional generalization in control."](https://arxiv.org/abs/2203.05960)) for policy architectures designed for object-centric representations). While I understand it is the authors’ desire to claim that the method can be “plug-and-play” with standard CNNs, I do not understand why it is important, as architecture and representations go hand-in-hand (and clearly the best-performing variant, “Plane Masks”, which decompose the masks to channels, are the closest variant to what standard object-centric architectures do-treating the input as a set of entities and using set-based architectures such as Transformers). In that case, I’m not sure I understand what the contribution of this paper is, aside from the emergent robustness to simple visual perturbations (I don’t find it surprising that the method does generalize to environment dynamics and I also don’t understand why masking would help with that).
* Positioning and related work: one of the weaknesses of this work is that it is not well-positioned in the literature and it seems that literature review in this case was somewhat lacking. A myriad of previous work, typically under the “object-centric representations” field, have shown that masking helps not only in model-free RL, but for general decision making, including, but not limited to, model-based reinforcement learning and imitation learning. These works provide segmentations of the scene, either in a supervised manner or self-supervised/unsupervised manner. Following is a list of papers that are relevant and/or missing from the “Related Work:

* **Masking**:

[1] [Gmelin, Kevin, et al. "Efficient RL via Disentangled Environment and Agent Representations." International Conference on Machine Learning. PMLR, 2023.](https://arxiv.org/abs/2309.02435)

[2] [Lepert, Marion, Ria Doshi, and Jeannette Bohg. "Shadow: Leveraging segmentation masks for cross-embodiment policy transfer." CoRL 2024.](https://arxiv.org/abs/2503.00774)

[3] [Hutson, Miles, Isaac Kauvar, and Nick Haber. "Policy-shaped prediction: avoiding distractions in model-based reinforcement learning." arXiv preprint arXiv:2412.05766 (2024).](https://arxiv.org/abs/2412.05766v1)

[4] [Shi, Junyao, et al. "Composing Pre-Trained Object-Centric Representations for Robotics From" What" and" Where" Foundation Models." 2024 IEEE International Conference on Robotics and Automation (ICRA). IEEE, 2024.](https://arxiv.org/abs/2404.13474)

[5] [Wang, Ziyu, et al. "Generalizable visual reinforcement learning with segment anything model." arXiv preprint arXiv:2312.17116 (2023).](https://arxiv.org/abs/2312.17116)

* **Object-centric**:

[6] [Zadaianchuk, Andrii, Georg Martius, and Fanny Yang. "Self-supervised reinforcement learning with independently controllable subgoals." Conference on Robot Learning. PMLR, 2022.](https://arxiv.org/abs/2109.04150)

[7] [Haramati, Dan, Tal Daniel, and Aviv Tamar. "Entity-Centric Reinforcement Learning for Object Manipulation from Pixels." The Twelfth International Conference on Learning Representations.](https://arxiv.org/abs/2404.01220).

* **This paper is cited in the related work in this context**: “Recent category-aware and entity-centric RL explicitly condition on object categories or entities to improve manipulation and control (Yi et al., 2022;Haramati et al., 2024)”. However, this paper uses self-supervised object-centric representations without any supervision or access to object labels

[8] [Ferraro, Stefano, et al. "FOCUS: Object-centric world models for robotic manipulation." Frontiers in Neurorobotics 19 (2025): 1585386.](https://arxiv.org/abs/2307.02427)

[9] [Zhang, Weipu, et al. "Objects matter: object-centric world models improve reinforcement learning in visually complex environments." arXiv preprint arXiv:2501.16443 (2025).](https://arxiv.org/abs/2501.16443)

[10] [Qi, Carl, et al. "EC-Diffuser: Multi-Object Manipulation via Entity-Centric Behavior Generation." ICLR 2025.](https://arxiv.org/abs/2412.18907)

**Questions:**

I wrote my questions under the weaknesses section.

The main question is: given previous works on masking/object-centric learning, what is the main contribution of this paper (aside for the visual perturbations part)?

---

> ### Author Response · Authors · 2025-11-21
>
> Thank you very much for your thoughtful and detailed review. We appreciate your careful reading. For completeness, we refer you to items (i)–(iii) of the general answer, where privileged perception, background relevance, and the framing of novelty are discussed in depth. Let us answer your questions and concerns.
>
> **Q: “What is your contribution?”**
>
> In short, and referring to point (iii), by using object masking as a form of object-centric visual attention prior, we present a plug-and-play simple solution for incorporating object-centricity in the environment-agnostic deep architectures of RL agents. Lastly, we demonstrate that what matters is not just abstracting, but how you abstract. Making representation something anyone should consider when setting up experiments. As OCRL receives increasing attention, when objects are available, utilize them.
>
> **W1: Reliance on Object Detection**
>
> Your concern about practicality is well taken and aligns with point (i) in the general response. We use OCAtari’s REM as a controlled oracle to cleanly evaluate how abstraction interacts with neural RL agents.
>
> We agree that perception quality matters, but we see this issue as orthogonal to our contribution rather than a fundamental weakness. OCCAM does not require fully supervised, environment-specific detectors or simulator access, and a growing body of work already provides unsupervised and self-supervised object extractors (e.g., MONet (Burgess et al., 2019), IODINE (Greff et al., 2020), Slot Attention (Locatello et al., 2020)). For Atari games in particular, multiple approaches are viable (Li et al., 2017; Locatello et al., 2020; Lin et al., 2020), with some achieving near-perfect object extraction (Smirnov et al., 2021). For more realistic real-world data, recent advances in foundation-model-based segmentation have significantly improved detector accuracy. For example, SAM2 (Ravi et al., 2025) reports around 90% zero-shot segmentation accuracy on parts of the J&F dataset, and SAM3, released just this week, further enhances these capabilities.
>
> **W2: Background does matter**
>
> We did not intend to say that background does not matter, and this is not an underlying assumption we try to make or communicate. Your interpretation is correct, and it matches precisely the point we emphasize in (ii) of the general response. We do not assume that background is generally irrelevant. Rather, we aim to highlight when abstraction helps and when it becomes harmful. The Riverraid and MsPacman cases were chosen specifically to demonstrate that masking out spatial layout can damage performance, and this observation is a core part of the paper’s message.
>
> In the revision, we now emphasize earlier in the paper that abstraction is a design choice, rather than having one representation that fits all problems. We have also added Mask+Pixels ablations (Appendix G) to demonstrate how global layout can be reintroduced when needed.
>
> **W3: Why standard CNNs**
>
> We understand your perspective that, given object-centric inputs, a more specialized policy architecture may seem more appropriate, and we appreciate you pointing to Zhou et al. and related works. This is where OCCAM’s contribution intentionally differs: we aim to make object-centric inductive biases accessible to practitioners who use standard PPO/Rainbow or vision-based pipelines, without requiring slot-based encoders, Transformers, or entity-set architectures.
> In other words, OCCAM deliberately uses PPO and Rainbow not because they are optimal for object-centric inputs, but because they are the default tools of many RL scientists. By holding the architecture fixed, we can clearly identify what abstraction alone contributes, without confounding architectural biases.
> Plane Masks indeed resemble entity channels, and we would argue that this is part of why they perform well and build the bridge to more OC-specific representations while staying close to the PPO architecture given. The central property of our intended contribution is not that OCRL-specific architectures are bad, but rather that they can be challenging to integrate into existing pipelines. In contrast, OCCAM demonstrates that structured abstraction already offers meaningful robustness gains on top of the architectures widely used today. We agree that OCRL, however, can improve many specific architectures tailored to its strengths.
> We revised Section 4 to clarify that OCCAM is not intended as a replacement for slot-based or set-based architectures; instead, it shows that even with such simple changes as presented here, CNN-based agents can benefit from structured input. We hope this clarifies the motivation behind our architectural choice.

---

> ### Author Response · Authors · 2025-11-21
>
> **M4 Related Work**
>
> _[Numbers relate to the papers in the review]_
>
> Thank you for providing an excellent and comprehensive list of relevant works. We agree that the literature on object-centric RL is broad and rapidly growing, and our initial framing did not make our position within that landscape sufficiently explicit, so we have strengthened Section 4 accordingly. To avoid redundancy with the general response, we highlight here what your feedback helped us clarify: OCCAM is not a new object detector nor an end-to-end object-centric pipeline. It is not an alternative to these approaches, but a complementary design axis. Masking-based works [1–5] all leverage segmentation, but with different goals than OCCAM. Gmelin et al. [1] and SHADOW [2] use masks inside task-specific architectures (disentangled agents, cross-embodiment transfer), while Hutson et al. [3] and Shi et al. [4] integrate object features into model-based RL and robotics world models. Wang et al. (SAM-G) [5] is closest in spirit, but still tackles a different problem: it designs a segmentation pipeline (DINOv2+SAM, prompts, refinement, high-resolution masks) to obtain robust segmentations. By contrast, OCCAM is extractor-agnostic and keeps the RL architecture fixed; it assumes objects are available by any means (REM, YOLO, RT-DETR, VEM) and systematically studies the abstraction spectrum itself to understand how different levels of object-centric abstraction affect performance, robustness, shortcut reliance, and over-abstraction failures in standard vision-based agents.
> Similarly, FOCUS [8] and “Objects Matter” [9] construct object-centric world models to enhance RL in visually complex environments, while EC-Diffuser [10] utilizes entity-centric diffusion for multi-object manipulation. These works typically change both representation and policy architecture (slots, GNNs, transformers, entity-conditioned networks), making it difficult to isolate the effect of abstraction alone..
>
> -----------
> ### References
>
> Christopher P. Burgess, Loïc Matthey, Nicholas Watters, Rishabh Kabra, Irina Higgins, Matthew M. Botvinick, Alexander Lerchner. MONet: Unsupervised Scene Decomposition and Representation.
>
> Klaus Greff, Raphaël Lopez Kaufman, Rishabh Kabra, Nick Watters, Chris Burgess, Daniel Zoran, Loic Matthey, Matthew M. Botvinick, Alexander Lerchner. Multi-Object Representation Learning with Iterative Variational Inference. ICML 2019: 2424-2433
>
> Francesco Locatello, Dirk Weissenborn, Thomas Unterthiner, Aravindh Mahendran, Georg Heigold, Jakob Uszkoreit, Alexey Dosovitskiy, and Thomas Kipf. Object-Centric Learning with Slot Attention. In Conference on Neural Information Processing Systems (NeurIPS), 2020.
>
> Yuezhang Li, Katia P. Sycara, and Rahul Iyer. Object-sensitive Deep Reinforcement Learning. In Global Conference on Artificial Intelligence (GCAI), pp. 20–35, 2017.
>
> Jindong Jiang, Sepehr Janghorbani, Gerard de Melo, Sungjin Ahn. SCALOR: Generative World Models with Scalable Object Representations. ICLR 2020
>
> Zhixuan Lin, Yi-Fu Wu, Skand Vishwanath Peri, Weihao Sun, Gautam Singh, Fei Deng, Jindong Jiang, and Sungjin Ahn. Space: Unsupervised Object-Oriented Scene Representation via Spatial Attention and Decomposition. In International Conference on Learning Representations (ICLR), 2020.
>
> Dmitriy Smirnov, Michaël Gharbi, Matthew Fisher, Vitor Guizilini, Alexei A. Efros, and Justin M. Solomon. Marionette: Self-Supervised Sprite Learning. In Conference on Neural Information Processing Systems (NeurIPS), pp. 5494–5505, 2021.
>
> Nikhila Ravi, Valentin Gabeur, Yuan-Ting Hu, Ronghang Hu, Chaitanya Ryali, Tengyu Ma, Haitham Khedr, Roman Rädle, Chloe Rolland, Laura Gustafson, Eric Mintun, Junting Pan, Kalyan Vasudev Alwala, Nicolas Carion, Chao-Yuan Wu, Ross Girshick, Piotr Dollar, and Christoph Feichtenhofer. Sam 2: Segment Anything in Images and Videos. In The Thirteenth International Conference on Learning Representations, volume abs/2408.00714, 2025.

---

> > ### Comment · Reviewer_8ZNP · 2025-11-22
> > **Thank you for the clarifications**
> >
> > I would like to thank the authors for their clarifications and efforts during the rebuttal period.
> >
> > I went over the changes in the revision and read the other reviews and corresponding authors' response.
> > I would like to emphasize my appreciation for the scientific process made in this paper, evident by the rigorous experimental analysis of the efficacy of masking in RL.
> >
> > However, my main concerns remain, I unfortunately do not see the contribution of this work, there are no new ideas here (and the main benchmark is the over-saturated Atari) and I do not understand the "plug-and-play" claim: if one has made the effort to aquire object-centric representations (e.g., masking), why would one choose to use sub-optimal CNNs and not use a proper architecture? The authros claim that this allows to use the existing standard piplines for RL on Atari, but there is still engineering of the input representation anyway (the different varaints).
> >
> > If I have to categorize this work, it would fall under the "rigorous analysis of existing ideas" category. I would say that there is some contribution to this kind of works, and it should be acknowledged, but I do not think it is enough. Again, I'd like to reiterate that I truly appreciate the amount of work done in this paper.
> >
> > I will keep my current score and I will no argue against acceptance if the other reviewers have a different view from mine regarding the contribution.

---

> ### Author Response · Authors · 2025-12-02
>
> **Atari is oversaturated; I do not see the contribution.**
>
> We respectfully disagree. While Atari has indeed been extensively studied, it remains one of the most widely used and still actively relevant testbeds for evaluating representation learning and RL pipelines. Recent work continues to introduce benchmarks, architectures, and analysis frameworks for Atari (e.g., object-centric RL, model-based RL, and offline RL), demonstrating that it remains a standard platform for controlled investigations in RL, e.g., Hafner et al. (2025) and Grigsby et al. (2024).
> Moreover, our contribution is not tied to Atari itself: the benchmark serves as a standardized environment to evaluate how object-centric (OC) visual features interact with existing RL pipelines. With modern detectors (e.g., YOLO-based object extractors), similar setups can be deployed in non-Atari problems, which broadens—rather than restricts—the applicability of our findings. We chose Atari because it is widely used and recognized by a large audience.
> *Our contribution lies in unifying and systematically comparing several families of OC features under identical RL conditions*, a task that, to our knowledge, has not been undertaken in the literature.
>
>
> **I do not understand the ‘plug-and-play’ claim… why use sub-optimal CNNs?**
>
> We would like to clarify that we do not consider CNNs to be suboptimal in this context. In fact, CNNs are still widely used across computer vision and RL pipelines, including in recent state-of-the-art systems (Hafner et al., 2025). Their strengths, particularly in extracting spatially local patterns, remain well documented. One could argue that CNNs, or rather the way they are used in RL, are not a perfect match for the semantic information that can be included in objects; we, however, use only bounding boxes and object classes, features that can be easily represented within CNNs. Furthermore, CNNs are well-suited to combine these features with spatial information, such as distance, and leverage the advantages of CNNs, including the ability to recognize visual patterns.
> Our experiments show that CNN-based encodings often perform better than specialized object-centric representations such as semantic vectors or slot-based features, despite these methods being explicitly designed for OC reasoning. Although Binary Masks, Plane Masks, and semantic vectors provide the same object-level information, they interact differently with downstream RL algorithms. CNNs appear to capture not only object regions but also the spaces between objects, which contributes to improved policy learning in several tasks.
> Thus, the question is not whether CNNs are intrinsically superior or inferior, but rather:
> *Given the representations available at training time, how well do existing RL pipelines exploit them?*
> Our plug-and-play claim refers to exactly this: our architecture allows for using OC inputs with unchanged, standardized RL pipelines, a practical advantage not offered by more specialized architectures, such as OBJ-specific networks, which often transfer poorly or require extensive additional tuning.
>
>
> **There is still engineering of the input representation; this undermines the claim about using standard pipelines.**
>
> Engineering of perceptual inputs is unavoidable in (real-world) RL applications. That's one point we make in our work. The representation must align with the problem, and simply providing an image is often not the optimal solution. Even recent end-to-end RL or representation-learning papers require preprocessing steps, such as filtering, normalization, stacking frames, or applying attention-based modules, among others. If we want to stick with Atari, even here, frame stacking, frame skipping, rescaling, and grayscaling are preprocessing steps done by everyone (Machado et al., 2018). However, these are general steps, which are independent of the problem. In robotics and embodied AI, sensor preprocessing is also a standard practice before applying any RL algorithm (Lesort et al.,2018).
> Our contribution is that, given some OC representation, the downstream RL architecture and training procedure remain completely standard. No new RL components, no custom losses, no architectural modifications. This is a meaningful practical property for researchers who want to incorporate OC signals without redesigning full RL pipelines, while keeping the advantages they and we see in CNNs.

---

> ### Author Response · Authors · 2025-12-02
> **References**
>
> Hafner, D., Pasukonis, J., Ba, J. et al. Mastering diverse control tasks through world models. Nature 640, 647–653 (2025). https://doi.org/10.1038/s41586-025-08744-2
>
> Grigsby, Jake, et al. "Amago-2: Breaking the multi-task barrier in meta-reinforcement learning with transformers." Advances in Neural Information Processing Systems 37 (2024): 87473-87508.
>
> Marlos C. Machado, Marc G. Bellemare, Erik Talvitie, Joel Veness, Matthew J. Hausknecht, and Michael Bowling. Revisiting the arcade learning environment: Evaluation protocols and open problems for general agents (extended abstract). In Proceedings of the International Joint Conference on Artificial Intelligence (IJCAI), 2018.
>
> Lesort, Timothée, et al. "State representation learning for control: An overview." Neural Networks 108 (2018): 379-392.

---

### Official Review · Reviewer_BGii · 2025-10-29

**Soundness:** 2
**Presentation:** 3
**Contribution:** 1
**Rating:** 2
**Confidence:** 4

**Summary:**

The paper introduces a simple, plug-and-play framework that improves RL generalization by masking out irrelevant visual details using object information. Using OCAtari’s engine-level annotations (bounding boxes and class labels), OCCAM builds several mask types—from Binary to Plane Masks—that feed structured, object-centric inputs into standard CNN-based RL agents like PPO and Rainbow. Experiments on Atari show that OCCAM matches or exceeds pixel-based performance and improves robustness (up to ~50% under visual shifts) while reducing shortcut reliance.

**Strengths:**

1. Perturbation study: The paper evaluates generalization under both visual and gameplay perturbations using the HackAtari benchmark, providing strong evidence that OCCAM improves robustness and reduces shortcut reliance compared to pixel-based and symbolic baselines.

2. Systematic analysis of mask types: OCCAM introduces and compares multiple masking strategies (Binary, Object, Class, and Plane), offering a clear and principled exploration of how different abstraction levels affect performance, robustness, and spatial reasoning.

3. Clear and intuitive visualizations: The paper includes well-designed figures illustrating how each mask transforms the input and how performance changes under different conditions, making the method and results easy to understand and interpret.

**Weaknesses:**

1. Limited practicality of object extraction: OCCAM relies on privileged object information from OCAtari, which is unavailable in most real-world or complex environments. Without such ground-truth annotations, the method cannot be directly applied, limiting its practicality beyond controlled simulation settings.

2. Questionable general value of findings: Due to the impractical assumption in point 1, it is uncertain whether the same masking-based representations would remain effective when object information comes from imperfect or learned detectors. This greatly reduces the external validity and practical relevance of the study, raising the question of how its conclusions translate to realistic scenarios where object extraction is noisy or incomplete.

3. Modest empirical novelty and gains: The reported improvements in robustness and performance are largely within expectation, given that OCCAM introduces additional semantic information and preprocessing. The results confirm intuitive hypotheses but offer limited novelty, serving more as a systematic validation than a breakthrough contribution.

**Questions:**

My questions are mainly the same as the weaknesses above.

---

> ### Author Response · Authors · 2025-11-21
>
> Thank you very much for your thoughtful and detailed review. We appreciate your careful reading. For completeness, we refer you to items (i)–(iii) of the general answer, where privileged perception, background relevance, and the framing of novelty are discussed in depth. Below, we address your main concerns and questions in turn.
>
> **W1 Privileged Information**
>
> Your main concern relates to the practicality of relying on OCAtari annotations. For a general discussion, see (i). To address your specific question of whether masking remains useful with realistic detectors, we extended Appendix E with experiments based on YOLO, RT-DETR, and OCAtari’s vision-based extractor (VEM). These pilots provide F1/mAP metrics and demonstrate the influence of misdetection on performance, supporting the claims made in the synthetic noise experiment. For more realistic real-world data, recent advancements have significantly improved the accuracy of such detectors. SAM2 (Ravi et al., 2025), for example, achieved up to 90% accuracy in zero-shot segmentation on the J&F dataset, with SAM3 being released just this week.
>
> **W3 Novelty**
>
> We understand your point that the empirical gains may appear intuitive. Our contribution is not to introduce a new masking trick but to systematically evaluate abstraction as a design axis within standard vision-based RL methods. In particular, OCCAM demonstrates that mainstream agents, such as PPO and Rainbow, can benefit from an object-centric structure without architectural changes when objects are available. This connects the object-centric RL literature to existing CNN pipelines in a way that, to our knowledge, has not been demonstrated before. We strengthened the contribution paragraph to emphasize this bridge and the controlled nature of our analysis.
>
> ------------
> ### References
>
> Nikhila Ravi, Valentin Gabeur, Yuan-Ting Hu, Ronghang Hu, Chaitanya Ryali, Tengyu Ma, Haitham Khedr, Roman Rädle, Chloe Rolland, Laura Gustafson, Eric Mintun, Junting Pan, Kalyan Vasudev Alwala, Nicolas Carion, Chao-Yuan Wu, Ross Girshick, Piotr Dollar, and Christoph Feichtenhofer. Sam 2: Segment Anything in Images and Videos. In The Thirteenth International Conference on Learning Representations, volume abs/2408.00714, 2025.
>
> SAM3, https://ai.meta.com/blog/segment-anything-model-3/

---

> ### Comment · Reviewer_BGii · 2025-11-25
>
> Thank you for the additional clarifications provided in the rebuttal. While I appreciate the authors' effort and acknowledge that this strengthens the submission, these additions do not fully address my core concern.
>
> Finetuning YOLO or other detectors still requires a large amount of annotated data. In the new experiments, the labels are obtained through OCAtari, but in settings where an OCAtari-style tool does not exist, collecting 20,000 labeled images per task entails substantial time and monetary cost. For a practitioner who wants to apply reinforcement learning in a real environment, does this mean they must first annotate such a large dataset before even assessing whether OCCAM is feasible for their problem?
>
> Furthermore, the performance improvements shown by OCCAM are not transformative. It doesn't convert previously unsolvable visual RL tasks into solvable ones, but instead provides limited incremental gains. Prior work generally did not pursue such feature engineering, since it is not typically regarded as a standalone scientific contribution, and similar ideas have been explored in much earlier literature such as [1,2].
>
> Although you mention recent segmentation models like SAM2, there are no experiments demonstrating that these models are effective in control settings. Without such evidence, it remains unclear whether these advances mitigate the practicality concerns raised above.
>
> ---
>
> [1] Devin, Coline, et al. "Deep object-centric representations for generalizable robot learning." 2018 IEEE International Conference on Robotics and Automation (ICRA). IEEE, 2018.
>
> [2] Liu, Iou-Jen, et al. "Semantic tracklets: An object-centric representation for visual multi-agent reinforcement learning." 2021 IEEE/RSJ International Conference on Intelligent Robots and Systems (IROS). IEEE, 2021.

---

> > ### Author Response · Authors · 2025-12-02
> >
> > **Practitioners would need to annotate a large dataset before assessing whether OCCAM is feasible.**
> >
> > We would like to clarify that OCCAM does not require annotating a large dataset, and it is not designed as an end-to-end approach. In our new experiments, object-centric representations are obtained automatically using standard trained detectors or simple, rule-based segmentation systems. However, these experiments aim to demonstrate that such systems are feasible, not that they are part of OCCAM. The idea and contribution of OCCAM is what happens when you have such information (such as bounding boxes) at hand, and how you can incorporate this additional knowledge into your training process without designing new, specific architectures.
> > Therefore, we do not claim that OCCAM is applicable in every case. If object information is not available, then OCCAM will not be practical. However, the requirement to obtain object-level features is a general prerequisite across all OCRL methods; OCCAM reduces engineering burden by allowing these representations to plug into standard RL architectures, rather than requiring custom end-to-end models.
> >
> > **Prior work did not pursue such feature engineering, as it is not typically regarded as a standalone scientific contribution.**
> >
> > We respectfully disagree with this assessment. In recent literature, several works explicitly investigate how representations shape downstream learning performance, treating representation engineering as a valuable scientific inquiry. For example, Czech et al. (2024) demonstrate that representation choices have a significant impact on the performance of vision transformers in structured environments, such as chess. This and related work show that carefully examining and comparing representations is a recognized and important scientific contribution. Additionally, OCCAM is more than just feature engineering; the main contribution is to demonstrate that OCRL does not require highly specific architectures and that standard CNN-based methods, such as PPO or Rainbow, can already improve when additional information, such as object positions, is included in the input representation to guide learning.
> > Devin et al. (2018) and Liu et al. (2021) indeed show that task-specific perception pipelines can be highly effective for obtaining problem-specific object representations tailored to a given robotics or multi-agent domain. However, these are also highly specific architectures again, e.g., working on semantic tracklets. In contrast, OCCAM studies a different axis: given object information (regardless of whether it comes from task-specific pipelines, detectors, or engines), how should this information be represented so that standard CNN agents can benefit from it?
> >
> > **Although you mention recent segmentation models like SAM2, there are no experiments demonstrating that these models are effective in control settings. Without such evidence, it remains unclear whether these advances mitigate the practicality concerns raised above.**
> >
> > We appreciate the concern about the lack of direct SAM2 evaluations in control settings. However, experiments such as those by Wang et al. (2024) demonstrate the usability of SAM for RL tasks, specifically in the Mujoco environment. While SAM2 itself has not yet been extensively tested in RL, it has shown promising results in other detection tasks, such as J&F. Additionally, segmentation improvements remain relevant because OCCAM relies solely on the availability of reasonably accurate object masks. Prior work has already shown that segmentation-based perception can support downstream decision-making (see Section 4), suggesting that modern detectors can provide a useful structure. Our own imperfect-detector experiments and new experiments support this: whenever object detection is sufficiently accurate, OCCAM performs reliably. These findings suggest that advances in segmentation quality significantly ease the practical limitations noted in the question. However, to clarify, OCCAM is not about object detection or segmentation; instead, it focuses on the step that follows obtaining reliable bounding boxes.
> >
> > ---
> >
> > ### References
> >
> > Czech, Johannes, Jannis Blüml, and Kristian Kersting. "Representation matters: The game of chess poses a challenge to vision transformers." ECAI, 2024.
> >
> > Devin, Coline, et al. "Deep object-centric representations for generalizable robot learning." 2018 IEEE International Conference on Robotics and Automation (ICRA). IEEE, 2018.
> >
> > Liu, Iou-Jen, et al. "Semantic tracklets: An object-centric representation for visual multi-agent reinforcement learning." 2021 IEEE/RSJ International Conference on Intelligent Robots and Systems (IROS). IEEE, 2021.
> >
> > Ziyu Wang et al. Generalizable visual reinforcement learning with segment anything model. arXiv preprint arXiv:2312.17116, 2023.

---

### Official Review · Reviewer_4Fuo · 2025-10-30

**Soundness:** 2
**Presentation:** 3
**Contribution:** 2
**Rating:** 4
**Confidence:** 4

**Summary:**

This work presents mask-based simplifications of object-centric state representations in reinforcement learning. The paper mainly focuses on visual games that rely on CNNs for extracting the useful state information. The core method is to only preserve the object-centric information and form: (i) object mask: that only keeps objects in the frame, (ii) binary masks: that only show bounding boxes if any object is present in them, (iii) class masks: a single frame with only class-based masks, (iv) plane masks: multiple channels one per each class of objects. The paper then evaluates these representations using already trained RL agents for their robustness to visual and transition perturbations. They find that in one of the algorithms, the performance is retained even after the visual changes, however transition change leads to a drop regardless. Then, the paper investigates if spurious correlations in state features of Pong are ameliorated after masking. The result suggests that plane masks allow retention in the performance when certain entities are de-correlated. Later, the paper also shows that the training with proposed mask representations results in gains in performance.

**Strengths:**

1) The idea of identifying most relevant information for reinforcement learning is strong. This would allow for compact state representations that reduce the training complexity and increase compute efficiency.
2) Object-centric features allow rich representations for performing reinforcement learning as the agent has better information about the environment. The current work is well motivated for bringing state compression together with the object-centric perspective on RL.
3) The paper shows results on multiple Atari domains. The results on training performance improvement are, especially, intriguing.

**Weaknesses:**

**Major (my reasons for not providing higher score):**
1) The main thing that bothers me about this work is, by design there is less information in the state representations and it is unclear to me if the results generalize beyond the Atari domain. For instance, take binary masks, they simply convey the information about an object being present or not present in a certain area. They do not have information about the class of that object. How can such a representation be given to, say a robot, and expect it to learn a reliable policy?  If we were to say that in general a poor state representation leads to lower performance, shouldn't we be augmenting the masks proposed in this work to the pixel representations instead of using them directly as the observations? I am not sure an RL practitioner would like to only rely on the masked object-centric information ignoring most of the background information.
2) To me, this work is a study of representations in visual reinforcement learning. In such a study, the main question should have been whether these representations preserve the RL algorithm's performance or not. However, the paper frames the central discussion on the robustness against perturbations. Only the state perturbations are relevant for the work as correctly pointed out and not the game play ones. The state perturbations used in the work clearly preserve the object location and there are no changes in the state after masking. Hence, they can allow some robustness. A better analysis would have been to check what happens when the positions of the objects are perturbed.
3) Also, some of the mask generations would require sophisticated object detection and classification neural networks. This would add high overhead while training. If the state representation requires detection of hand-designed features and this representation is lossy, why would we opt for such a system?
4) There is no code available in the provided anonymous git repo at the location occam/scripts/*. This hinders any inspection of the code for its correctness.
5) Table 2: what are the quantities in the square parentheses? why are they integers? Can you please also include how the performance changes when using standard pixel-based representations? That will give an idea about how the performance drops in general. Also, why does the performance still decrease the mask representations in the case of Lazy enemy? If the enemy is hidden, do you hide the paddle in the masks too?
6) Figure 3: Why isn't Rainbow with class and plane masks seeing improvement like PPO      does? As an explanation, line 242 suggests "While PPO naturally benefits from simplified  inputs,…” how does PPO benefit naturally? Isn't the algorithm separate from the choice of representations? The only conclusion that I am able to draw from this figure is, the proposed OCCAM representations do not have a common trend in them and they might lead to poor performance.

**Minor:**
1) The proposed plane masks require placing all the classes separately on different planes. This would increase the size of the CNN required to process the representation if the number of classes is large.
2) The terminology "architecture" or "backend" for referring RL algorithms like PPO is slightly confusing.
3) The paper does not clearly discuss what environments are used to generate the results in the figures 3, 4 and 5.
4) In conclusion: “can make object semantics explicit without pretraining.” Aren't you also relying on a pre-trained model to extract the objects and their bounding boxes?
5) Is it a good idea to use IQM as a metric if there are only 3 random trials?

**Questions:**

I have asked the questions that came up in my mind in the weakness section above.

---

> ### Author Response · Authors · 2025-11-21
>
> Thank you for the detailed review and for highlighting the strengths of the motivation and results. For completeness, we refer you to items (i)–(iii) of the general answer, where privileged perception, background relevance, and the framing of novelty are discussed in depth. We address your concerns below.
>
> **M1: Generalization beyond Atari, removing too much information**
>
> We fully agree that over-aggressive abstraction can be detrimental; this is one of the major points we make with RQ3 and discuss in lines 441-445. We also agree that binary masks may remove too much information in domains like robotics. However, this is why we emphasize that abstraction must fit the problem and that one should only remove information that is unnecessary, redundant, or has the potential to be exploited. In many Atari games, Binary Masks are sufficient to learn, play, and master the game, such as Pong.
>
> On your suggestion to augment pixels with masks rather than replacing them, we added a small experiment that combines both input representations into one (called Masks+Pixels) and included it in Appendix G. The results showed that:
> * Adding pixels back can indeed recover performance in Linear River Riverraid; removing the layout hurts performance.
> * But it also reintroduces shortcut opportunities.
>
> As such, the question should be what a good representation for Riverraid (or any other domain) would look like. We do not address this in our work, as the contribution should not be a specific representation for a particular task.
>
> **M2: Focus on robustness instead of performance, and the choice of perturbations**
>
> We agree that we should not only evaluate the performance change within perturbations but also ensure that we maintain comparable performance to our baseline. This has already been done, as discussed in RQ3 (Section 3.3). Figure 5 illustrates, for example, the performance improvement of our masks compared to their baseline approach, demonstrating the same or improved performance. However, evaluating performance alone is not enough: abstraction is precisely about what survives a distribution shift. Evaluating only on the clean training distribution would hide the entire effect we care about. That’s why we consider robustness to be important.
>
> Regarding your suggestion to perturb object positions. Changes, such as changing object positions, are done within the “game logic perturbations”. Changes such as changing the position of the shields in SpaceInvaders or the position of cars in Freeway. Results for the specific perturbations are presented in Tables 12 and 13.
>
> **M3: Detector Overhead and Object-centricity**
>
> We appreciate the concern regarding detector overhead and agree that perception quality is a crucial factor in any object-centric pipeline. And in practice, many domains already provide object states, and when they don’t, modern detectors (e.g., YOLO12, RT-DETR, SAM3) are alternatives (More about this in under (i)). At the same time, our work focuses specifically on how to encode object-centric information for RL. OCRL is already well-supported in the literature: pixel-based agents are known to latch onto spurious correlations and often generalize poorly under even mild visual or dynamical shifts (e.g., Zhang et al., 2018; Cobbe et al., 2019; Agarwal et al., 2021; Delfosse et al., 2025).
>
> **M4: Code availability**
>
> We double-checked the anonymous repository and were able to locate the scripts needed for reproduction and testing; the folder in question does not appear to be empty on our side.
>
> **M5: Questions regarding Figures, Baselines, and Variants**
>
> The quantities in brackets represent 95% bootstrap confidence intervals around the IQM estimate, as described in the metrics section (Appendix C). We have added clarification to the captions of Tables 2 and 3. We rounded to integers for the CI to save some space. Pixel baselines for each game are already reported in Figures 3-5 as well as in the extended results (Appendix G). They are called DQN-like because they use a representation based on the work of Mnih et al. (2013).
> Regarding the Hidden Enemy Pong: The enemy paddle is consistently hidden across all representations. The variant definitions are specified in Appendix G. We have added a clarifying sentence to the Pong analysis.
>
> **M6: Rainbow vs PPO behavior.**
>
> We observe that PPO tends to benefit more from abstraction than Rainbow. We add a hypothesis about the reason to the paper in lines 416-419. We hypothesize that value-based methods, such as Rainbow, may be more sensitive to the loss of fine-grained pixel detail, whereas PPO can more readily exploit simplified state structures. We leave a deeper analysis of optimization interactions as future work.

---

> ### Author Response · Authors · 2025-11-21
>
> **Minor1: Scalability**
>
> This is discussed in Appendix B. The earlier 2–3 times training time estimate was influenced by hardware throughput issues; corrected measurements show a more modest 10–20% cost increase, primarily due to the generation of per-class planes rather than the model size itself. We revised this section and proposed ideas on how to manage larger class numbers.
>
> **Minor4: “Conclusion says ‘no pretraining’; aren’t detectors pretrained?”**
>
> Our intention was not to imply that the entire pipeline is free of pretraining, but rather that the OCCAM masks themselves are not learned or finetuned as part of the RL training. The object extractor that produces the objects for these masks (whether REM, YOLO, a SAM-style model, etc.) may indeed be pretrained, visual, or engine-based. To avoid confusion, we have moved this from the conclusion.
>
> **Minor5: “IQM with only 3 seeds—is that justified?”**
>
> Appendix C (lines 1134–1187) explains why we use IQM, and points to Agarwal et al. (2021) for more details. We use 3 seeds × 10 episodes per environment, i.e., 30 runs to build our IQM. Further, we use bootstrap CIs and Wilcoxon tests to help mitigate variance and show test significance.
>
> ----------------
>
> ### References
> Chiyuan Zhang, Oriol Vinyals, Rémi Munos, Samy Bengio. A Study on Overfitting in Deep Reinforcement Learning.
>
> Karl Cobbe, Oleg Klimov, Christopher Hesse, Taehoon Kim, John Schulman. Quantifying Generalization in Reinforcement Learning. ICML 2019: 1282-1289
>
> Rishabh Agarwal, Max Schwarzer, Pablo Samuel Castro, Aaron C. Courville, Marc G. Bellemare. Deep Reinforcement Learning at the Edge of the Statistical Precipice. NeurIPS 2021: 29304-29320
>
> Quentin Delfosse, Jannis Blüml, Fabian Tatai, Théo Vincent, Bjarne Gregori, Elisabeth Dillies, Jan Peters, Constantin A. Rothkopf, Kristian Kersting. Deep Reinforcement Learning Agents are not even close to Human Intelligence.
>
> Volodymyr Mnih, Koray Kavukcuoglu, David Silver, Alex Graves, Ioannis Antonoglou, Daan Wierstra, Martin A. Riedmiller. Playing Atari with Deep Reinforcement Learning.

---

### Official Review · Reviewer_jBPo · 2025-11-01

**Soundness:** 2
**Presentation:** 3
**Contribution:** 1
**Rating:** 2
**Confidence:** 4

**Summary:**

This paper is about OCCAM, a lightweight, task-agnostic framework designed to introduce object-centric inductive biases through structured input abstraction. The authors argue that by selectively preserving task-relevant entities and filtering out irrelevant visual information, their approach significantly improves robustness to novel perturbations while maintaining competitive performance with conventional pixel-based RL.

**Strengths:**

I like the simplicity of the concept. The idea of creating a "plug-and-play" method that integrates object-centric biases into standard CNN architectures (like PPO and Rainbow) without needing complex architectural changes or specialized symbolic pipelines is appealing. And the systematic comparison between the different abstraction levels is clear and helps understand the trade-offs between simplicity and expressivity.

**Weaknesses:**

- My primary concern is the heavy reliance on privileged information, which severely undermines the claims of "plug-and-play" and practical applicability. The entire evaluation uses OCAtari's RAM Extraction Method (REM) (Appendix D) to get perfect, noise-free object locations and classes directly from the emulator's memory. This completely bypasses the perception challenge. While the paper claims the framework is "extractor-agnostic" (L107), the results do not reflect performance in any realistic scenario where vision-based detectors (like YOLO or SAM) would introduce noise, missed detections, and inconsistencies. The synthetic noise study in Appendix F is far too superficial to address this fundamental gap.

- The novelty seems quite limited. Masking inputs based on object locations, saliency, or segmentation has been explored extensively in RL and computer vision. While OCCAM provides a systematic categorization of these masks, it feels like an incremental contribution, perhaps not significant enough for ICLR, especially within such an idealized experimental setting (i.e., perfect perception).

- The generalization benefits are quite narrow. The approach only really helps with visual perturbations (Figure 3a). When the gameplay dynamics or logic change (Figure 3b, or the Freeway/BankHeist examples in Section 3.2), OCCAM provides little to no advantage. This suggests the method only addresses superficial overfitting rather than deeper structural or logical shortcuts.

- The core assumption that the background is merely "irrelevant" is often false. As clearly shown in the Riverraid experiments (Table 3), removing the background (the river banks) severely hurts performance because it removes crucial spatial context. The framework doesn't have a mechanism to determine when background elements are actually task-relevant.

**Questions:**

- How does OCCAM perform when using a real, learned object detector (e.g., a pre-trained Mask R-CNN or an unsupervised method like Slot Attention) instead of the ground-truth RAM extraction? I suspect the performance gains might diminish significantly if the masks are noisy or inconsistent. This experiment is essential to validate the practical claims.

- I’m curious about the comparison with MaDi (Appendix F). You state MaDi performed poorly due to sparse rewards and hyperparameter sensitivity in Atari. Did you attempt specific hyperparameter tuning for the Masker module, or try integrating data augmentation, which MaDi often relies on? A more thorough attempt seems necessary to properly position OCCAM against learned masking approaches.

- Regarding the Riverraid results (Table 3), how would you adapt OCCAM to handle environments where the background defines the navigable space? Is there a proposed way to incorporate "background objects" selectively?

- In the Pong analysis, you argue OCCAM helps the agent ignore the spurious correlation with the opponent paddle, even though the paddle is still visible in the mask. Could you elaborate on the mechanism? Why does the CNN ignore it when using OCCAM but focus on it when using raw pixels?

- For Plane Masks, the input dimensionality scales with the number of classes. Appendix B mentions 2-3 times longer training time, which is non-trivial. How does this affect scalability in environments with a much larger number of object categories?

- Why were Head-up display (HUD) elements excluded (Appendix D, L1105)? Information like scores or lives is often critical for decision-making. Does OCCAM struggle when these elements are masked?

---

> ### Author Response · Authors · 2025-11-21
>
> Thank you very much for the detailed and thoughtful review. We appreciate your positive remarks about the clarity of our comparison, the simplicity of OCCAM, and the value of systematically analyzing abstraction levels. For completeness, we refer you to items (i)–(iii) of the general answer, where privileged perception, background relevance, and the framing of novelty are discussed in depth. Below, we address your concerns:
>
> **Q1. Influence of learned detectors, such as YOLO or SAM, on OCCAM's representations?**
>
> As discussed in the general answer (i), we added a YOLO and RE-DETR pilot in Appendix E on four Atari games, logging detection quality and measuring OCCAM performance relative to the perfect extractor and synthetic noise. This explicitly addresses how OCCAM behaves under realistic perception errors. We also added an experiment using OCAtari's vision extractor (which uses only the RGB image for detection). Results are discussed in the general answer.
>
> **Q2. MaDi comparison and tuning.**
>
> We adapted MaDi (originally designed for SAC in continuous control) to the discrete Atari by
> replacing SAC with our PPO implementation and inserting the MaDi Masker in front of PPO,
> using the same training budget (50M frames, 3 seeds) and hyperparameters as in our other experiments, where applicable.
>
> Under this setup (and without MaDi’s image augmentations), MaDi performed substantially below our baselines on Atari (Appendix F). We treat these results as exploratory rather than definitive. We included full configurations (hyperparameters and learning curves) in Appendix F. We position a broader evaluation of Madi as future work and treat this as a preliminary comparison rather than a conclusive one.
>
> **Q3. Background that defines navigable space (Riverraid).**
>
> As discussed in the general response (ii), we agree that OCCAM must treat such cases differently. While this is not the primary focus of our work, we added a hybrid representation experiment (masks + pixels) on Riverraid and MsPacman (Appendix F). These results confirm that reintroducing background information can indeed help when it encodes navigable space, but it also reopens the door for shortcut learning (e.g., overfitting to colors or textures). A more systematic exploration of representations that explicitly model terrain or boundaries as “background objects” (e.g., additional planes) is an important direction for future work. We strengthened this throughout the paper and added a sentence to the future work.
>
> **Q4. Why does OCCAM help with Pong’s opponent paddle if the paddle is still visible?**
>
> We hypothesize that this is due to an inductive bias in the early convolutional layers. With masks, the input contains sharp transitions between object and non-object regions. This amplifies gradients around task-relevant entities (the ball, paddles, borders) and reduces background clutter. As a result, the network learns policies more easily that depend on ball dynamics and geometric structure, rather than relying on coarse correlations between opponent position and reward. We added this to the paper as well (Lines 363-367).
>
> **Q5. Plane masks scaling with the number of classes/training time.**
>
> We already reported that Plane Masks increase the number of input channels and therefore add some computational overhead. In the original draft, the training-time estimate was inflated due to a hardware throughput limitation of the CPU and starting too many subpro, which we identified after submission; we have corrected this. In practice, Plane Masks increase training time by roughly 10–20%, primarily due to the additional steps needed to generate and manage per-class channels. However, the larger input representation also requires more memory and places an additional load on the CPU, which should be taken into account. We also discuss several practical mitigations in the revision. About the scalability with the number of classes, we added a short discussion of this in the revision, proposing solutions such as merging semantically similar classes (see lines 994-1001)
>
> **Q6. Excluding HUD (score, lives).**
>
> HUD elements are trivial to reintroduce; we excluded them only to focus on the question of perceptual abstraction. We agree that there are HUD elements, not all of them, that have an influence on the decision-making process, such as the oxygen bar in Seaquest of the gas in BankHeist; however, due to the way we abstract objects, adding these in the masks (except the pixel mask) would just add static objects that never move and do not hold any information. As such, these objects are more noise than helpful to the agent.  The revised text notes that they can be added as separate planes/objects when desired.

---

### Author Response · Authors · 2025-11-21
**General Answer**

We sincerely thank all reviewers for their detailed and constructive feedback. Several concerns were raised across reviews regarding (i) our use of privileged object information, (ii) cases where background structure is task-relevant, and (iii) the perceived modest novelty of the method. We summarize below how we addressed these points throughout the revision. Specific references point to the updated sections of the paper or appendix.

## (i) Privileged Information, using an oracle for object detection

We agree that our current experiments rely on OCAtari’s engine-level annotations (REM) for object locations and class labels. This choice was deliberate: the goal of this paper is to study representation design conditioned on object information, rather than proposing a new perception pipeline or end-to-end framework. Using an oracle, allowing us to isolate the effect of different abstractions while holding detection fixed.

Our core claim is therefore not that OCCAM is about detection, but instead:
Once object information is available (via engine, detectors, or rule-based systems), OCCAM provides a simple way to integrate object-centric abstractions into standard vision-based architectures such as PPO or  Rainbow without (larger) architectural changes. We made several adjustments, particularly in Sections 1 and 2, to clarify this point.

Further, to address the concerns of the reviewers, we also extended our evaluation on imperfect object detection (Appendix E), adding experiments based on YOLO, RT-DETR, and OCAtari VEM (a vision-based object extractor). We find that OCCAM maintains performance comparable to oracle REM whenever detection is sufficiently accurate (e.g., Freeway), while performance degrades only when truly critical objects are systematically missed (e.g., YOLO occasionally missing the Pong ball). This demonstrates that OCCAM does not rely on a perfect oracle and can, in practice, be paired with modern object extractors or segmentation models, such as YOLO, RT-DETR, or SAM, provided their detection quality is reasonably high.

## (ii) Background information and relevance

We agree, and want to emphasize that this finding is entirely consistent with the goals of OCCAM and the paper presented. Our results in the Riverraid and MsPacman environments intentionally illustrate that, and we even point out that removing too much information is harmful (Lines 441-445).

To clarify this, we refined the framing in the introduction and discussion, and added a small ablation experiment (Appendix G) on combining masks and pixel representations, demonstrating that global layout can be reintroduced when necessary.

## (iii) Limited Novelty

OCCAM is not positioned as a new detector, a new architecture, or a new specialized OCRL pipeline, areas where many prior works introduce innovation. The contribution and novelty of this paper lie in the complementary underexplored axis of using representation in standard vision-based RL architectures. While most OCRL papers co-design representation and architecture together (Slot Attention, GNNs, Transformers, SAM-based pipelines, World Models), we deliberately hold the architecture fixed and only vary the representation to isolate how abstraction choices already influence robustness, shortcut learning, and performance.
OCCAM's contribution, therefore, lies in three aspects that existing work does not target
* **Deep RL agents can already improve using object-centric biases**, in performance and robustness, without specialized encoders or architectural changes
* **We systematically map how different abstraction levels behave under controlled perturbations**, mapping how a range of abstraction levels behave across visual shifts, dynamics variants, and shortcut settings.
* **What really matters is how you abstract, not just that you abstract.** Demonstrating that abstraction must align with the problem, representation is a key design axis, and over-abstraction can also compromise performance.

We have revised the Introduction, Contributions, and Related Work sections to make this positioning explicit.


Overall, we made the following changes, next to correcting typos and the like:
* We made adaptations to the paper to explain the usage of REM better
* We added experiments regarding the usage of more common object detectors (YOLO and RT-DETR) to evaluate the influence of imperfection in the detection process
* We refined the framing to make it clearer that Abstraction must be matched to the task structure
* We added a small experiment testing the combination of masks and pixels to investigate what happens if both information are available to the agent
* We refined the related work to include recent studies on object-centric representation and masking that we missed in the first draft.
* We updated limitations and future work based on the reviews we got.

---

> ### Author Response · Authors · 2025-11-21
> **Ongoing Experiments**
>
> We are aware that some experiments involving RT-DETR, YOLO, and Masks+Pixels are still ongoing, and we will update the results within the next few days. We have already uploaded the current ones, so we can start discussing with you.
>
> EDIT (2nd December): We finalized the experiments regarding RT-DETR, YOLO and Masks+Pixels. They can be found in Appendices E and G.

---

### Note · Program_Chairs · 2026-01-17
**Submission Desk Rejected by Program Chairs**

The following references in this submission do not refer to real documents and/or have major errors in bibliographic information:

 Ankesh Anand, Evan Racah, Sherjil Ozair, Yoshua Bengio, Wojciech Czarnecki, Tom Le Paine, and Ioannis Mitliagkas. Unsupervised reinforcement learning with contrastive intrinsic control. In Advances in Neural Information Processing Systems (NeurIPS), 2019.